# Acetylcholine modulates the precision of prediction error in the auditory cortex

David Pérez-González[1,2,3†], Ana Belén Lao-Rodríguez[1,2†], Cristian Aedo-Sánchez[1,2], Manuel S Malmierca[1,2,4*]

[1]Cognitive and Auditory Neuroscience Laboratory, Institute of Neuroscience of Castilla y León, Calle Pintor Fernando Gallego, Salamanca, Spain; [2]Institute for Biomedical Research of Salamanca (IBSAL), Salamanca, Spain; [3]Department of Basic Psychology, Psychobiology and Behavioural Science Methodology, Faculty of Psychology, Campus Ciudad Jardín, University of Salamanca, Salamanca, Spain; [4]Department of Biology and Pathology, Faculty of Medicine, Campus Miguel de Unamuno, University of Salamanca, Salamanca, Spain

*For correspondence:
msm@usal.es

†These authors contributed equally to this work

Competing interest: The authors declare that no competing interests exist.

**Abstract** A fundamental property of sensory systems is their ability to detect novel stimuli in the ambient environment. The auditory brain contains neurons that decrease their response to repetitive sounds but increase their firing rate to novel or deviant stimuli; the difference between both responses is known as stimulus-specific adaptation or neuronal mismatch (nMM). Here, we tested the effect of microiontophoretic applications of ACh on the neuronal responses in the auditory cortex (AC) of anesthetized rats during an auditory oddball paradigm, including cascade controls. Results indicate that ACh modulates the nMM, affecting prediction error responses but not repetition suppression, and this effect is manifested predominantly in infragranular cortical layers. The differential effect of ACh on responses to standards, relative to deviants (in terms of averages and variances), was consistent with the representational sharpening that accompanies an increase in the precision of prediction errors. These findings suggest that ACh plays an important role in modulating prediction error signaling in the AC and gating the access of these signals to higher cognitive levels.

## eLife assessment

The findings of this study are **valuable** as they provide new insights into the role of acetylcholine in modulating sensory processing in the auditory cortex. This paper reports a systematic measurement of cell activity in the auditory cortex before and after the microiontophoretic application of Ach during an oddball and cascade sequence of auditory stimuli. The evidence presented is **convincing**, as the study used a rigorous experimental design and statistical analysis. The manuscript will interest researchers in auditory neuroscience and neuromodulation, as well as clinicians and individuals with auditory processing disorders.

## Introduction

Neuromodulation strongly impacts sensory processing by influencing neuronal excitability or synaptic processes in neuronal circuits (*Hasselmo and Giocomo, 2006*; *Lucas-Meunier et al., 2003*; *Metherate, 2011*; *Muñoz and Rudy, 2014*; *Picciotto et al., 2012*). Acetylcholine (ACh) is a widely distributed neuromodulator throughout the brain, including the AC, and modulates processes such as attention, learning, memory, arousal, sleep, and/or cognitive reinforcement (*Batista-Brito et al., 2018*; *Dalley et al., 2004*; *Franklin and Frank, 2015*). The main source of ACh to the AC is the basal

forebrain (*Chavez and Zaborszky, 2017*; *Mesulam, 2013*; *Zaborszky et al., 2008*). In the auditory system, cholinergic modulation is known to alter frequency response areas generating changes across frequency tuning, decreasing the acoustic threshold at the characteristic frequency, and changing the encoding of the spectral representation of many auditory neurons (*Ma and Suga, 2005*; *Metherate, 2011*). Thus, ACh promotes neuronal and synaptic plasticity at different temporal scales (*Kamke et al., 2005*; *Kilgard and Merzenich, 1998*).

Here, we characterized the effect of ACh on stimulus-specific adaptation (SSA), a type of neuronal adaptation found in the AC. SSA is elicited by an oddball paradigm, which comprises a pattern of repeating sounds (standards), interrupted by a low-probability and unexpected sound (deviant). The deviant usually differs in frequency, but could differ on any other physical dimension from the standard sound, or otherwise violate a pattern of regularity established by the standard (*Chen et al., 2015*; *Nieto-Diego and Malmierca, 2016*; *Parras et al., 2017*; *Ulanovsky et al., 2003*; *von der Behrens et al., 2009*).

Thus, neurons that exhibit SSA adapt specifically to the standard stimulus but resume their firing when a deviant stimulus appears. SSA has been proposed to be a neuronal correlate of 'mismatch negativity (MMN)', an evoked potential obtained in human and animal electroencephalographic studies using the oddball paradigm (*Nieto-Diego and Malmierca, 2016*; *Ulanovsky et al., 2003*). Hence, SSA has been also referred to as nMM in animal models (*Casado-Román et al., 2020*; *Lao-Rodríguez et al., 2023*; *Parras et al., 2020*; *Parras et al., 2017*; *Malmierca et al., 2019*; *Carbajal and Malmierca, 2020*), a nomenclature that highlights deviance detection, as well as other underlying processes other than neuronal adaptation. Because nMM is considered a form of short-term plasticity (*Ogawa and Oka, 2015*) and ACh has been shown to play a role in this type of neural plasticity (*Marshall et al., 2016*; *Moran et al., 2013*; *Parr and Friston, 2017*; *Vossel et al., 2012*) it is plausible that ACh may be involved in the generation and/or modulation of nMM. Furthermore, it has been shown that ACh differentially modulates the neural response to the standard stimulus in neurons of the IC (*Ayala and Malmierca, 2015*).

Currently, MMN and nMM are best explained by the predictive coding theory (*Friston and Sporns, 2008*; *Friston, 2005*). According to the predictive coding framework, higher-level cortical areas generate predictions about sensory streams that are conveyed in a top-down manner to lower hierarchical levels, to suppress the ascending neuronal activity evoked by sensory events that can be anticipated. However, when current predictions do not match the sensory inputs, the lower levels then send forward bottom-up prediction errors to higher hierarchical levels (*Friston and Kiebel, 2009*).

In the setting of an oddball paradigm, every evoked response is considered to be a prediction error response. When the stimulus does not change, sensory learning enables better predictions to progressively attenuate the response to standards until an oddball or deviant stimulus is encountered — and a large prediction error is seen. The prediction error response to oddball stimuli may be further augmented by increasing the precision or gain of prediction errors: predictive coding models weigh sensory prediction errors by their precision, which is the inverse of the variance. Precision weighting of prediction errors, therefore, encodes the confidence in the accuracy of the information (i.e. prediction errors) in terms of their signal-to-noise ratio (*Parr and Friston, 2017*). In neurobiological terms, precision has been suggested to be mediated by synaptic gain modulation; namely, the sensitivity of postsynaptic responses to neural inputs (*Adams et al., 2022*), and is likely implemented by cholinergic neuromodulation (*Moran et al., 2013*; *Yu and Dayan, 2005*; *Yu and Dayan, 2002*). Increasing the gain of certain neuronal populations — e.g., superficial pyramidal cells reporting prediction errors – affords them privileged access to higher levels of processing and ensuing Bayesian belief updating. This can be likened to attentional gain in psychological terms – or increasing the Kalman gain in terms of Bayesian filtering.

According to the predictive coding model, there are, therefore, two mechanisms underlying the MMN/nMM (*Auksztulewicz and Friston, 2016*; *Carbajal and Malmierca, 2018*; *Friston and Sporns, 2008*; *Harms et al., 2021*; *Parras et al., 2017*). First, nMM could reflect the repetition suppression (RS) of the response to the predictable stimuli (standards) due to sensory learning – that resolves sensory prediction errors as predictions become more accurate. In addition, nMM could also reflect an increase in the precision or gain afforded prediction errors, when the (oddball) stimulus cannot be predicted; enabling the ascending prediction errors to revise higher-level representations about the auditory stream. In short, oddball stimuli elicit an enhanced neural response when an unexpected

(deviant) stimulus is presented. Repetition suppression and prediction error modulation can be distinguished using control sequences (*Ruhnau et al., 2012*), and there is now evidence in both humans and rodents that MMN/nMM receives contributions from both prediction error (modulation) and repetition suppression (sensory learning) at various levels of the auditory system (*Ishishita et al., 2019*; *Parras et al., 2017*).

The main goal of the present study was to determine the role that ACh plays in the modulation of prediction errors and repetition suppression in the rat AC; utilizing control sequences to disambiguate these mechanisms (*Nieto-Diego and Malmierca, 2016*; *Parras et al., 2017*). We used microiontophoretic injections of ACh to determine how ACh neurotransmission affects the responses of AC neurons that exhibit nMM, and to establish the role of ACh on prediction error modulation and repetition suppression, in both primary and secondary AC areas.

## Results

To explore the influence of cholinergic modulation on nMM, we recorded a total of 113 units in the AC before, during and after the microiontophoretic injection of acetylcholine in the rat. Recording depths ranged from 120–1191 μm including neurons from all layers.

To allocate each recorded neuron to a specific field in the AC, we recorded the frequency response area (FRA) and analyzed the topographical distribution of CF (characteristic frequency) for each unit. Each recording was assigned to a dorsoventral and rostrocaudal coordinate system relative to bregma as in previous studies (*Nieto-Diego and Malmierca, 2016*; *Parras et al., 2017*; *Polley et al., 2007*). This analysis allowed us to pool the data from all animals (*Figure 1a*) and construct a synthetic map of the CF across the entire rat auditory cortex (*Nieto-Diego and Malmierca, 2016*; *Parras et al., 2017*). Similar to these previous studies, we found a high-frequency reversal zone between ventral auditory field (VAF, caudally) and anterior auditory field (AAF, rostrally), a low-frequency reversal zone between A1 and posterior auditory field (PAF, dorso-caudally), and a high-frequency reversal between VAF and supra-rhinal auditory field (SRAF, ventrally). Thus, we could reliably define the lemniscal (A1, AAF, and VAF) and non-lemniscal (SRAF, PAF) auditory cortical fields as shown in *Figure 1b*.

### ACh modulates the magnitude of neuronal responses and nMM

After the FRA was identified, we selected a pair of pure tones (10–30 dB above the minimum threshold) within the FRA at each recording site to test the adaptation of the neuronal response under the oddball paradigm. To do so we quantified nMM using the common SSA index (CSI). CSI values may range from –1 to +1, with negative values for units that respond more to standard stimuli and positive values for units that respond more to deviant stimuli (see Methods). In the following, we describe the effect of ACh on nMM (*Figure 2*; *Figure 3*; *Figure 4*; *Figure 5*; *Figure 6*), in terms of prediction error modulation and repetition suppression (*Figure 7*). *Figure 2* illustrates representative responses of two units from the lemniscal auditory pathway — during and after the injection of ACh — that showed a change in the CSI during ACh release (*Figure 2a and b*, decreased and increased CSI, respectively). Similarly, the non-lemniscal, divisions (high-order regions) of the AC (*Figure 3*) also show decreased and increased CSI during ACh application (*Figure 3a and b*; respectively). In both *Figures 2 and 3*, peri-stimulus time histograms (PSTH) are also depicted for standard- and deviant stimuli (blue and red lines, respectively). On average, the firing rate decreased during the application of ACh (mean control: 5.8±7.0 spikes/s, mean ACh: 4.7±10.3 spikes/s; Wilcoxon signed-rank test, p<0.001). Most neurons recovered their basal firing rates after 60–90 min after the ACh injection (recovery mean: 6.5±8.1 spikes/s, Wilcoxon signed-rank test, p=0.4, control vs. recovery).

Given the changes in the FRAs observed in individual units during the application of ACh (*Figures 2 and 3*), we checked for bandwidth changes at the population level. For that, we measured the FRA bandwidth in octaves at 10 and 30 dB above the minimum threshold (BW10 and BW30, respectively). Numerous individual units showed either broadening or sharpening of their FRA bandwidth during ACh application, which were averaged out at the population level (*Figure 4a, b*). Indeed, no significant changes occur in neither BW10 (control: 0.76±0.57 oc taves; effect: 0.88±0.75 oc taves; Wilcoxon signed-rank test, p=0.248) nor BW30 (control: 1.33±0.73 oc taves; effect: 1.43±0.80 oc taves; Wilcoxon signed-rank test, p=0.664) (*Figure 4c*).

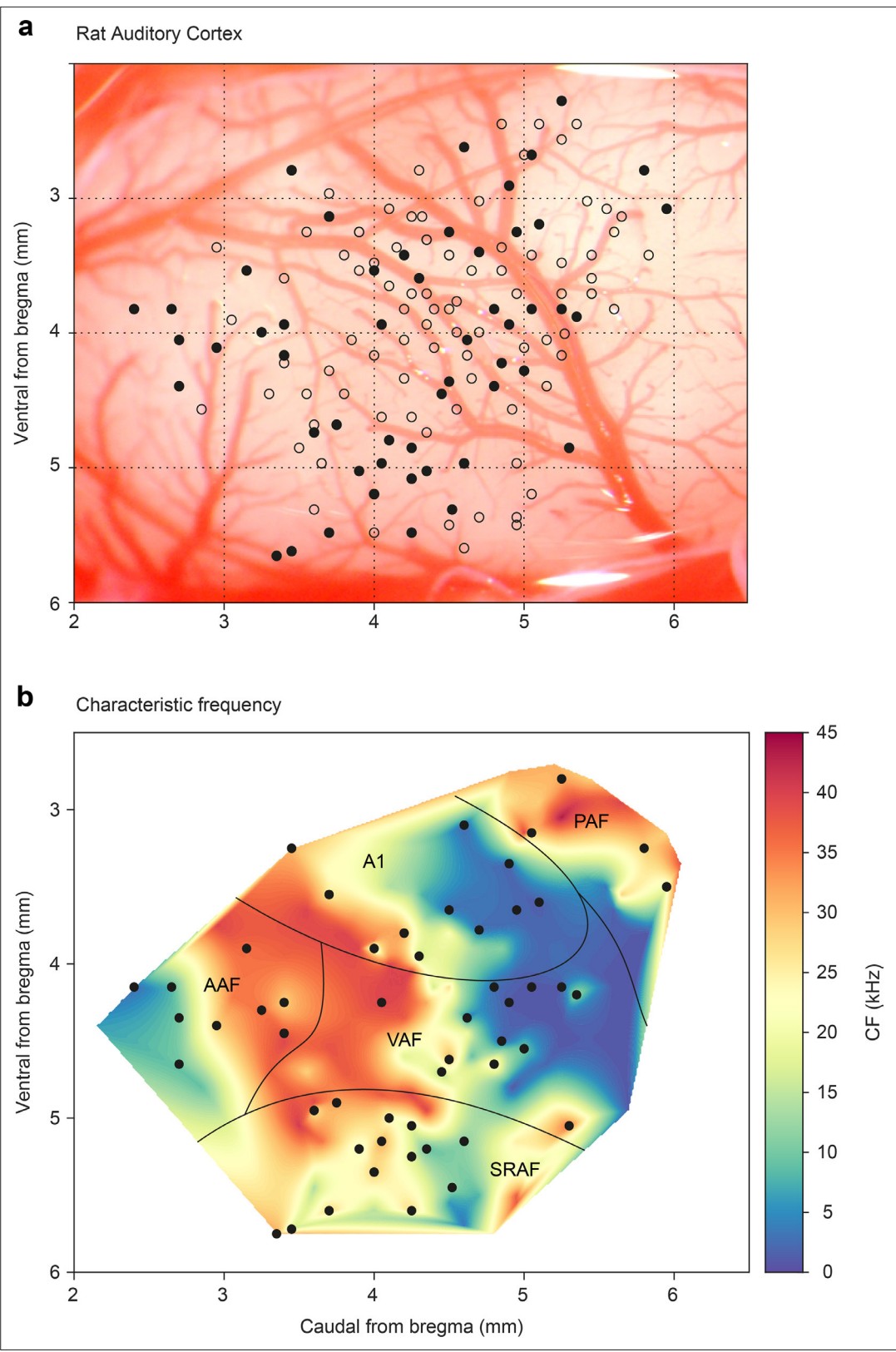

**Figure 1.** Map of all recording locations. (**a**) All recording sites are drawn over the cortex of a representative animal. At every site, the characteristic frequency (CF) was determined and then we presented an oddball paradigm and the corresponding control sequences. (**b**) Distribution of the CFs across the entire rat auditory

*Figure 1 continued on next page*

*Figure 1 continued*

cortex. Note how each field shows a characteristic CF gradient. Empty dots indicate sites used only for CF determination. Filled dots indicate sites where oddball paradigm was recorded.

The online version of this article includes the following source data for figure 1:

**Source data 1.** Source data for *Figure 1*.

Next, we analyzed the effect of ACh on the CSI at the population level (n=85; comprising the neurons in our sample with significant responses to both $f_1$ and $f_2$ auditory stimuli). ***Figures 2 and 3*** illustrate four examples of units from lemniscal and non-lemniscal subdivisions before, during and after the application of ACh. Modulation of CSI levels was observed in all individual examples. ACh produced a significant increase of the CSI value in 26% (***Figure 5a***, purple dots; 22/85, bootstrapping, 95% *c.i.*) and a significant decrease in 31% of the neurons recorded (***Figure 5a***, orange dots; 26/85, bootstrapping, 95% *c.i.*). Thirty-seven units (43%) were unaffected by ACh. The average CSI during ACh injection (CSI = 0.54±0.27) was similar to that of the control condition (CSI = 0.57±0.23; Wilcoxon signed-rank test, p=0.359; ***Figure 5b***).

We analyzed the effect of ACh on the response to the deviant- or standard stimuli, separately. We observed that ACh produced a significant decrease in the mean firing rate in response to the deviant tones (***Figure 5c***; 11.75±17.46 Hz), as compared to the control condition (16.51±16.19 Hz; Wilcoxon signed-rank test, p<0.001), as well as in the mean firing rate in response to standard tones (***Figure 5c***; 4.58±6.38 vs. 3.97±9.90 Hz, control vs ACh; Wilcoxon signed-rank test, p=0.001).

In order to establish how ACh affects discriminability of deviant- and standard stimuli — i.e., how well the stimulus identity can be decoded from the neuronal firing rate – we calculated the area under the curve (AUC) of the receiver operating characteristic (ROC) for each tone (***Figure 5d***, left panel) and the Mutual Information to determine how much the response distributions for deviant- and standard stimuli overlap before and after ACh injections and how well the stimulus identity can be decoded from the neuronal firing rate. AUC indicates how different the neuronal responses to deviant- and standard stimuli are, and thus, how well a neuron is able to discriminate between both types of stimuli, considering that the neuron would respond differently to them (in terms of firing rate). Completely overlapping firing rate distributions (in response to deviants and standards) would yield an AUC = 0.5, while separate distributions would result in AUC = 1. On the other hand, Mutual Information is defined as the reduction in the uncertainty of one variable (in this case the neuronal response) due to the knowledge of another (in this case, the type of stimulus; ***Timme and Lapish, 2018***). In other words, this measure reflects how much information a neural response (firing rate) contains about the type of stimulus that is played. In practical terms, high Mutual Information implies that a neuron is able to discriminate between stimuli, and thus respond differently to them. This analysis shows that the mean AUC decreased significantly during the application of ACh (0.74±0.13 vs. 0.67±0.15, control vs. effect; Wilcoxon signed-rank test, p<0.001). The same happened for Mutual Information (***Figure 5d***, right panel; 0.09±0.07 vs. 0.07±0.08, control vs. effect; Wilcoxon signed-rank test, p=0.003). These AUC and Mutual Information analyses indicate that under the influence of ACh, neurons are less able to discriminate deviant stimuli from standard stimuli.

## Different mechanisms for SI increase or decrease during ACh application

As mentioned above, the CSI is an indicator of the average level of nMM for a single unit, since it takes into account the responses of two frequencies ($f_1$ and $f_2$) that alternatively play the role of deviant and standard. However, in order to better understand the effect of ACh on the response to each stimulus class — depending on whether it was presented as a deviant or as a standard — the frequency-specific SSA index (SI, see Methods) is a better indicator. We calculated the SI for 205 individual frequencies, from 113 units that elicited a significant response to that particular frequency from our sample. This analysis shows that ACh produced a significant increase of the SI in 27% of the frequencies recorded (***Figure 6a***, purple dots; 55/205, bootstrapping, 95% *c.i.*), 31% (64/205; ***Figure 6a***, orange dots) showed a significant decrease of their SI and 42% frequencies (86/205) were unaffected by ACh. The mean SI (SI = 0.55±0.31, control condition) was reduced during ACh injection (SI = 0.40±0.50) although this decrease was not significant (Wilcoxon signed-rank test, p=0.086; ***Figure 6b***).

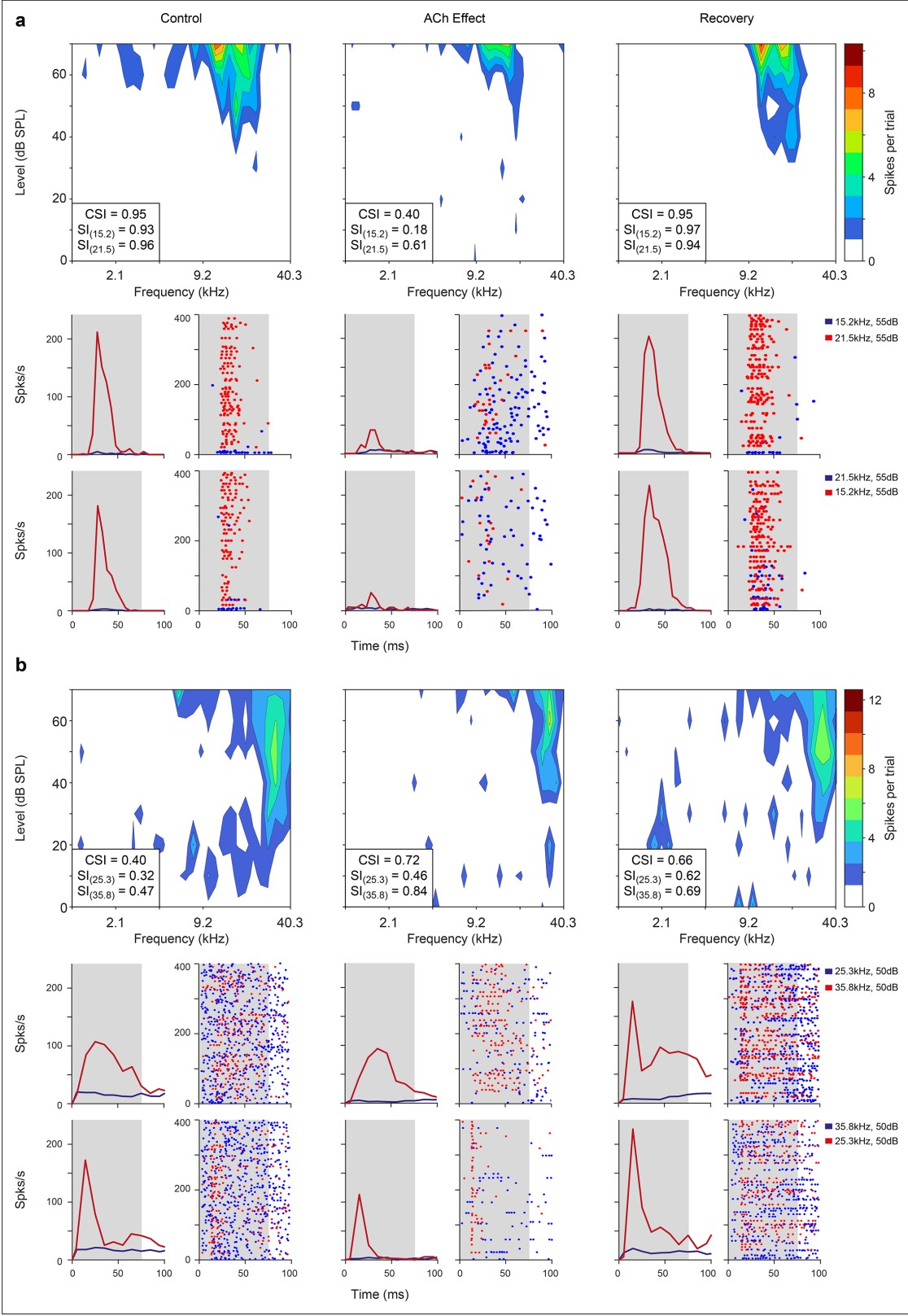

**Figure 2.** Examples of lemniscal auditory cortex (AC) units modulated by acetylcholine (ACh). Left, middle, and right columns show the responses before (control), during (effect), and after (recovery) ACh application, respectively. (**a**) Frequency response areas (FRAs) and peri-stimulus time histograms (PSTHs) from a unit located in A1. This specific neuron shows a common SSA index (CSI) decrease during ACh application and recovery to

*Figure 2 continued on next page*

*Figure 2 continued*

baseline levels (CSI = 0.95, CSI = 0.4, and CSI = 0.95, respectively) (**b**) FRAs and PSTHs from an AAF unit with increased CSI (CSI = 0.4, CSI = 0.72, and CSI = 0.66, respectively) during ACh application. Red lines and dots: deviant response; blue lines and dots: standard response.

The online version of this article includes the following source data for figure 2:

**Source data 1.** Source data for *Figure 2*.

Since we found that the application of ACh had divergent effects on the SI (increase or decrease), we further analyzed separately those two groups to identify different mechanisms. We examined the firing rates in response to deviant- and standard stimuli, separately for those frequencies showing a SI increase or decrease during the application of ACh. In both cases, responses to the deviant- and standard stimuli were affected differently, when measuring the firing rate change ratio (amount of change caused by ACh, relative to the baseline firing rate). In frequencies showing SI decrease, the effect of ACh was significantly different for the deviant- and standard stimuli (*Figure 6c*, top panel; p<0.001, Wilcoxon signed-rank test). Furthermore, this ratio was clearly negative for deviant sounds (p<0.001, Wilcoxon signed-rank test) and not significant for standards (p=0.256, Wilcoxon signed-rank test), indicating that ACh caused a reduction of the firing rate differentially during deviant stimuli. On the other hand, in frequencies showing SI increase the effect of ACh also had a distinct effect on the deviant- and standard stimuli (*Figure 6d*, top panel; p<0.001, Wilcoxon signed-rank test), but in this case, the effects were reverse: the change ratio was clearly negative for standard sounds (p=0.001, Wilcoxon signed-rank test) and not significant for deviant ones (p=0.379, Wilcoxon signed-rank test), hence ACh caused a reduction of the firing rate primarily during standard stimuli. In other words, while the effect of ACh was a reduction of the firing rate, in some frequencies it only affected responses to the deviant stimuli, thus decreasing SI (*Figure 6c*, top panel), while in others it only affected responses to the standard stimuli, resulting in an increment of the SI (*Figure 6d*, top panel).

Interestingly, stimuli discriminability was affected differently by ACh in both groups. In frequencies showing SI decrease (*Figure 6c*, bottom panel) both ROC AUC and Mutual Information decreased significantly during ACh application (p<0.001 in both cases, Wilcoxon signed-rank test), in line with the results obtained for the whole sample (compare to *Figure 5d*). However, in frequencies showing SI increase (*Figure 6d*, bottom panel) ROC AUC did not change significantly (p=0.339), while Mutual Information increased slightly (p=0.008). These results suggest that frequencies showing SI decrease are the ones mediating the main effects of ACh in the population.

## ACh affects neuronal mismatch, modulating prediction error responses but not repetition suppression

According to the predictive coding framework, nMM can be split into two functionally distinct components: prediction error modulation (i.e. an increased response to an oddball caused by the violation of a regularity) and repetition suppression (i.e. reduction in the response to a standard caused by sensory learning of a repeated stimulus). These differential effects can be quantified by comparing the responses to the oddball paradigm with a suitable control sequence (*Figure 7—figure supplement 1*). Therefore, to test if ACh modulates prediction error signaling in AC, we also recorded the cascade control sequence (for the corresponding oddball sequences) in 66 neurons of the sample, using an approach previously published (*Carbajal and Malmierca, 2018*; *Parras et al., 2017*; *Ruhnau et al., 2012*; *Valdés-Baizabal et al., 2019*). First, we calculated the normalized spike counts in response to the cascade control, deviant, and standard stimuli (CAS, DEV, and STD, respectively; see Methods). ACh led to a significant decrease in the DEV responses in those units where SI decreased (*Figure 7a*; Wilcoxon Signed Rank test, p<0.001) and to a significant increase in those units where SI increased (*Figure 7b*; p=0.011). The opposite effect occurs on STD stimuli, increasing in units where SI decreased (*Figure 7a*; p<0.001) and decreasing in units where SI increased (*Figure 7b*; p<0.001). While this trend is very clear, there were no significant changes in CAS (p=0.091 and p=0.072, for SI decrease and increase, respectively).

Using the normalized CAS, DEV, and STD responses, we computed the indexes iMM, iPE, and iRS before and after the ACh injection in SI decrease units (*Figure 7c*, iMM: 0.63±0.19 vs. 0.13±42; iPE: 0.29±0.27 to -0.05±0.38; iRS: 0.34±0.20 vs. 0.17±0.49; control vs. ACh, respectively) and SI increase units (*Figure 7d*, iMM: 0.47±0.31 vs. 0.67±0.22; iPE: 0.15±0.36 vs. 0.33±43; iRS: 0.32±0.30 vs.

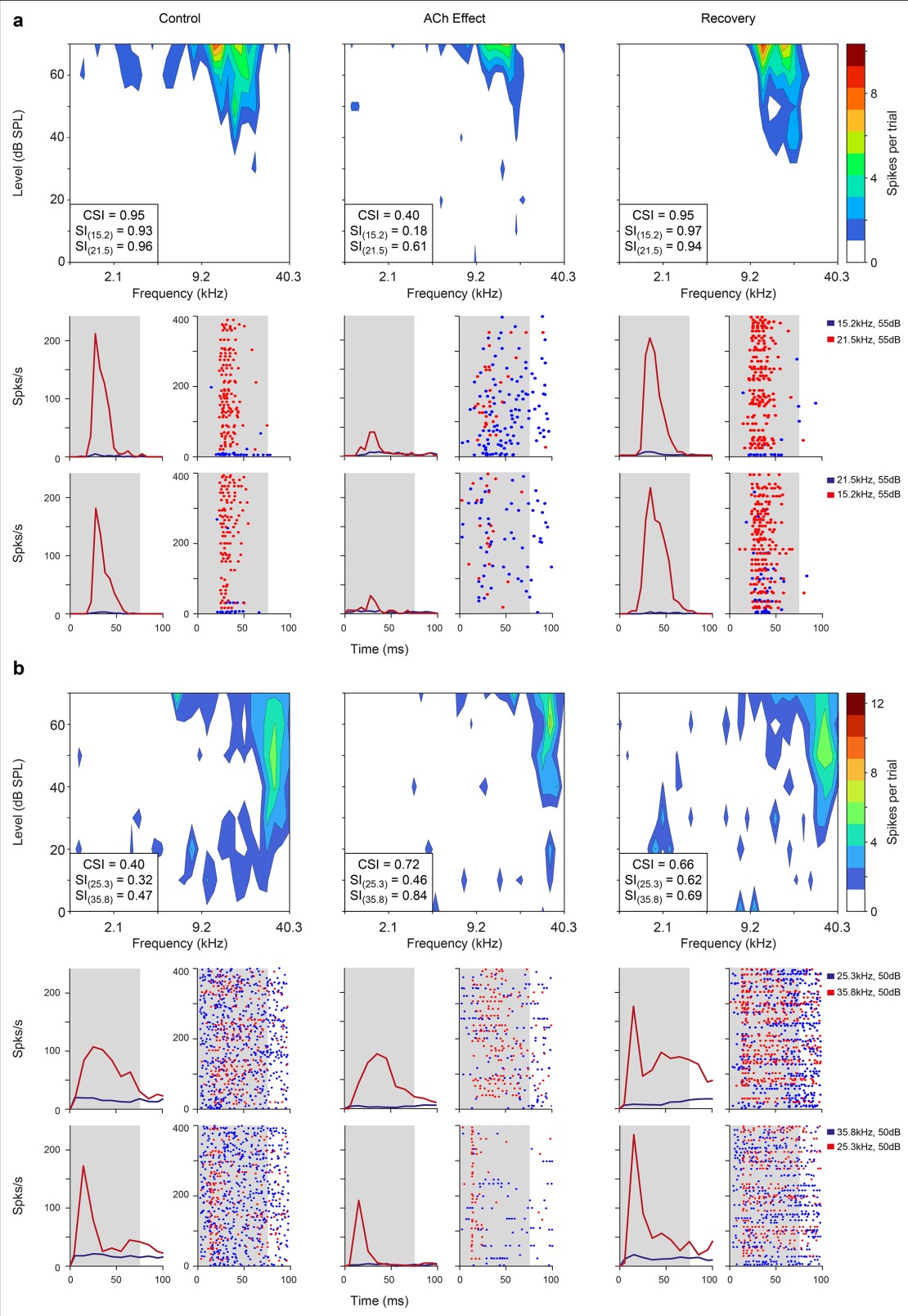

**Figure 3.** Non-lemniscal auditory cortex (AC) units modulated by acetylcholine (ACh). Left, middle, and right columns show the responses before (control), during (effect), and after (recovery) ACh application, respectively. (**a**) Frequency response areas (FRAs) and peri-stimulus time histograms (PSTHs) from a unit located in supra-rhinal auditory field (SRAF). This specific neuron shows a common SSA index (CSI) decrease during ACh application and recovery to baseline levels (CSI = 0.78, CSI = 0.19, and CSI = 0.77, respectively). (**b**) FRAs and PSTHs from a posterior auditory field (PAF) unit with

*Figure 3 continued*

increased CSI (CSI = 0.06, CSI = 0.5, and CSI = 0.11, respectively) during ACh application. Red lines and dots: deviant response; blue lines and dots: standard response.

The online version of this article includes the following source data for figure 3:

**Source data 1.** Source data for *Figure 3*.

0.34±0.31). iMM is equivalent to the CSI, but using normalized response values, and since it includes information about the cascade control, it can be divided into prediction error and repetition suppression components (iPE and iRS, respectively; see Methods and *Figure 7—figure supplement 1c*). Significant effects of ACh administration were found for iMM and iPE, both in SI decrease units (iPE: p<0.001; iMM: p<0.001; Wilcoxon signed-rank test; *Figure 7c*) and SI increase units (iPE: p=0.023; iMM: p<0.001; Wilcoxon signed-rank test; *Figure 7d*), while iRS remained unchanged (SI decrease units iRS: p=0.121; SI increase units iRS: p=0.675; Wilcoxon signed-rank test).

Under the predictive coding framework, the precision of prediction errors encodes the confidence or reliability afforded such errors (*Friston and Sporns, 2008*), enabling precise prediction errors to access higher levels of the auditory hierarchy. To test whether neuronal responses show differential effects of precision weighting — depending on their level of deviance sensitivity – we split our sample into two similar-sized groups (of frequencies), using the median SI as cutoff (0.6143: SI before the application of ACh): low SI (n=65 frequencies with cascade control) and high SI (n=66 frequencies with cascade control) (*Figure 8a–c*). Given this grouping, we found that ACh decreased significantly iMM (0.72±0.13 vs. 0.52±0.36, control vs. effect; p<0.001) and iPE (0.33±0.33 vs. 0.145±0.48, control vs. effect; p=0.004) in high SI frequencies, while only decreasing iRS significantly in low SI frequencies (0.28±0.25 vs. 0.18±0.36, control vs. effect; p<0.037) (*Figure 8a–c*, top row).

Given the divergent defects of ACh on putative prediction error modulation, we hypothesized that acetylcholine would increase the variance of deviance-sensitive responses over groups of cells. We calculated the variance of the above distributions of predictive coding indexes (iMM, iPE, and iRS) as an estimate of the (inverse of) their precision, as well as its 95% confidence interval using a 1000-sample bootstrap strategy, and then used an F-test to check for differences in the variances before and during ACh application (*Figure 8a–c*, bottom row). These results demonstrate that ACh application significantly increases the variance of iMM and iRS for both low SI and high SI groups, but interestingly ACh also significantly increases the variance of iPE only for the high SI group.

Then, we repeated the same variance analysis grouping the frequencies based on the anatomical layer location (depth) of the recording sites (*Figure 8d–f*). In this case, we split the frequencies into

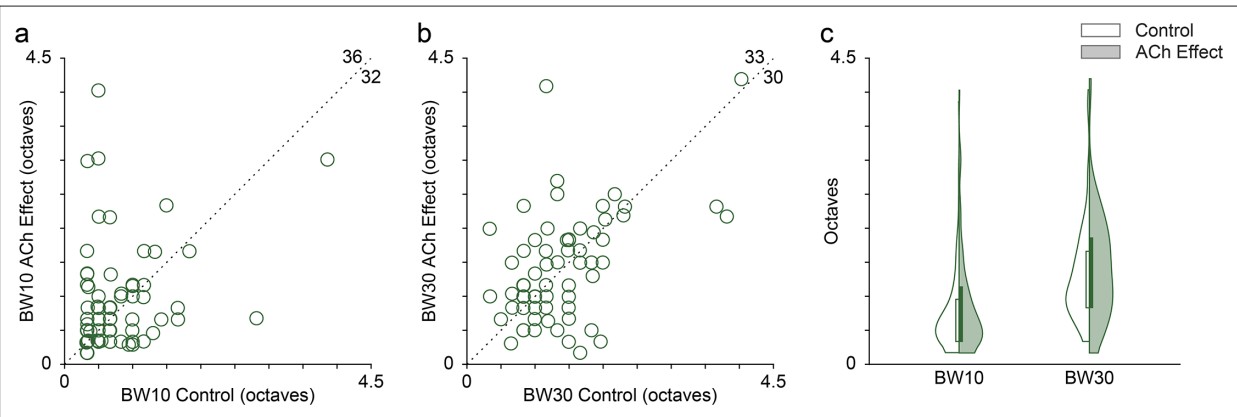

**Figure 4.** Effect of acetylcholine (ACh) on the bandwidth of the frequency response area (FRAs). Many individual units showed enlarged or reduced FRA bandwidths, both at 10 (**a**) and 30 (**b**) dB above the minimum threshold. The values in the upper-right corner of panels (**a**) and (**b**) indicate the number of units showing either an increment (top) or a decrement (right) of the bandwidth during the application of ACh. (**c**) Distribution of FRA bandwidths in octaves at 10 and 30 dB above the minimum threshold (BW10 and BW30, respectively), before (white background) and during (shaded background) the application of ACh. ACh did not alter significantly neither BW10 nor BW30 at the population level.

The online version of this article includes the following source data for figure 4:

**Source data 1.** Source data for *Figure 4*.

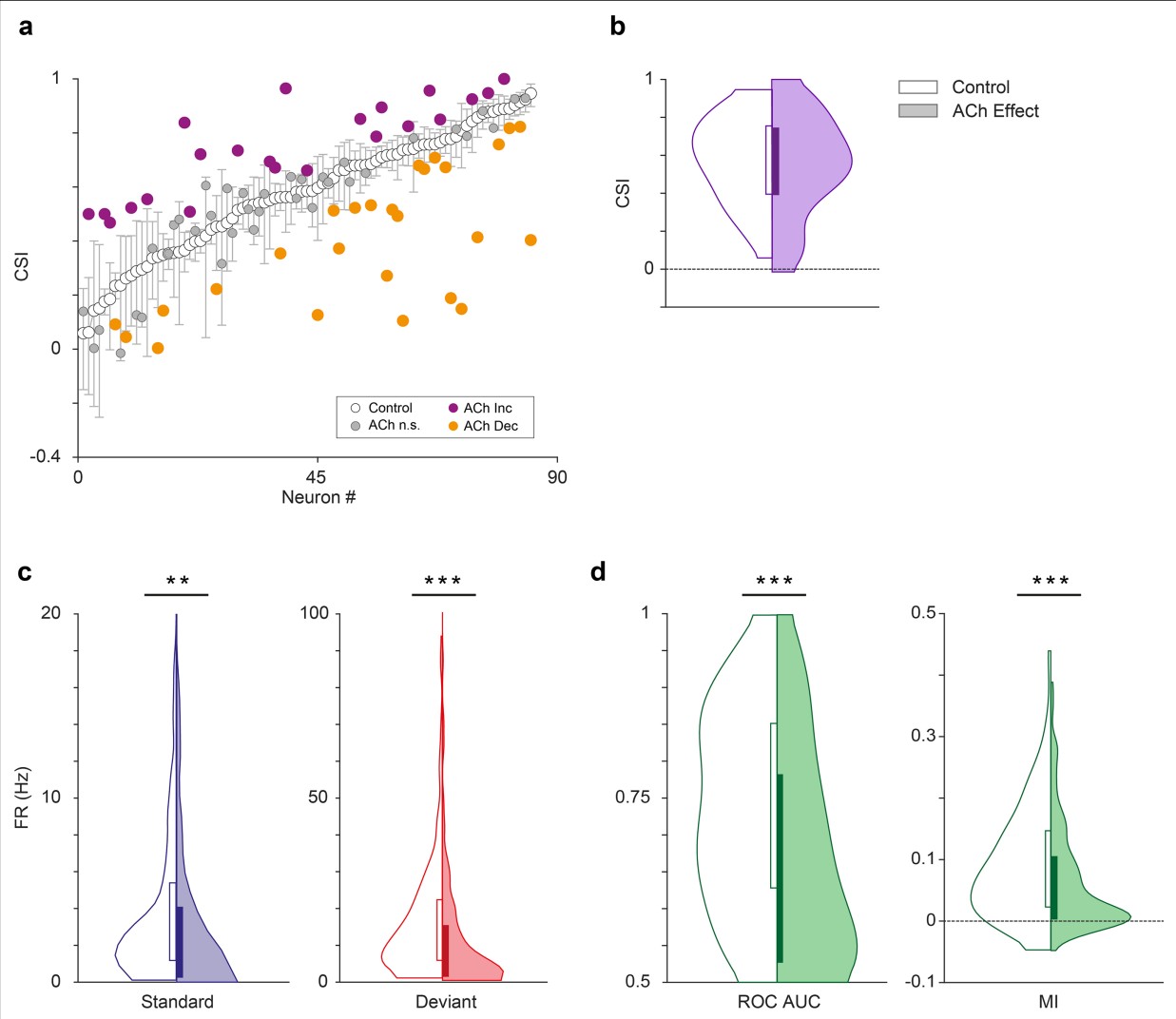

**Figure 5.** Effect of acetylcholine on common SSA index (CSI). (**a**) Changes on the CSI due to the effect of acetylcholine (ACh), for each recorded unit. The units are sorted based on their CSI in the control condition (white dots). The vertical bars indicate the 95% confidence interval for the CSI in the control condition. (**b**) Violin plots showing the CSI measured in control and effect conditions. The CSI did not change significantly during the application of ACh. In this and similar violin plots, white background indicates the control conditions, while shaded background indicates the effect condition. (**c**) Distribution of firing rates in response to the deviant (red) and standard tones (blue), in control and ACh conditions. The application of ACh caused a significant reduction in the response to both deviant and standard tones. (**d**) Distribution of area under the curve (AUC) and Mutual Information values in control and ACh conditions. The application of ACh caused a significant decrease in both AC and Mutual Information. **p<0.01; ***p<0.001.

The online version of this article includes the following source data for figure 5:

**Source data 1.** Source data for *Figure 5*.

deep and superficial, with a depth cutoff of 650 μm, which is near the boundary between layers IV and V in the rat AC (*Games and Winer, 1988*; *Malmierca, 2015*). In this case, we found no significant effect of ACh on iMM, iPE, nor iRS for none of the depth groups (*Figure 8d–f*, top row). However, the general trend was an increase in the variance for all indexes and depth groups during the application of ACh (except for iPE in superficial units), which was significant for all indexes in the case of deep units only (*Figure 8d–f*, top row).

Thus, our results indicate that ACh produces a general increase in the variance for iMM, iPE, and iRS. This general increase occurs regardless of the baseline SI value, but is more noticeable for neurons located in the infragranular layers.

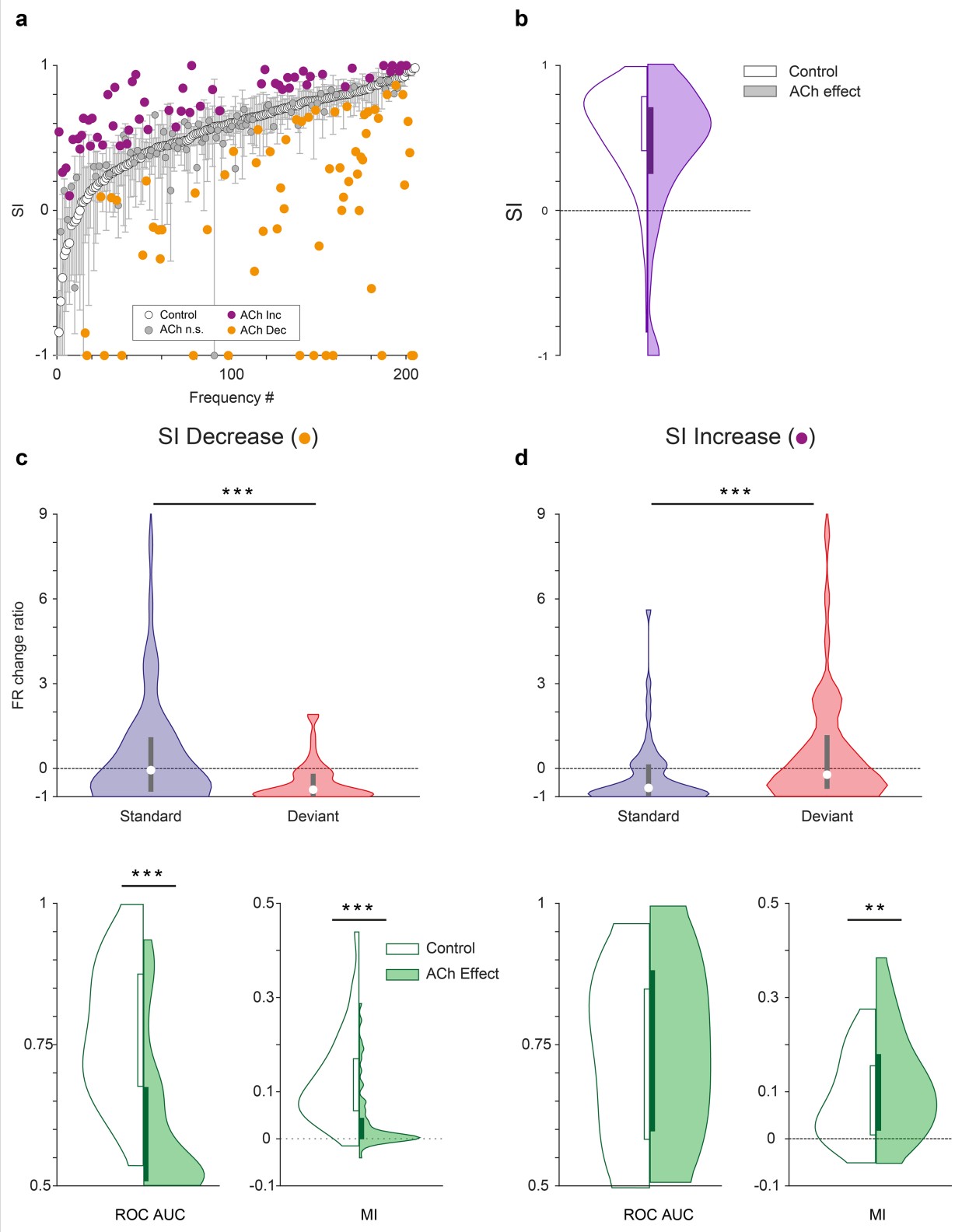

**Figure 6.** Characteristics of units depending on the effect of acetylcholine (ACh) on selectivity index (SI). (**a**) Changes on the SI due to the effect of ACh, for each tested tone. The units are sorted based on their SI in the control condition (white dots). The vertical bars indicate the 95% confidence interval for the SI in the control condition. (**b**) Violin plot depicting the overall effect of ACh on the SI, for the complete sample. (**c**) Firing rate change ratio due to ACh application, relative to the control condition. In units showing SI decrease (left column), it was caused by a significant reduction of the firing rate

*Figure 6 continued on next page*

*Figure 6 continued*

in response to deviant tones. By contrast, in units showing SI increase (right column) it was caused by a significant reduction of the firing rate in response to standard tones. (**d**) In units showing SI decrease (left column), there was a significant reduction of both area under the curve (AUC) and Mutual Information during the application of ACh. On the other hand, in units showing SI increase (right column), AUC did not change but Mutual Information increased. **p<0.01; ***p<0.001.

The online version of this article includes the following source data for figure 6:

**Source data 1.** Source data for *Figure 6*.

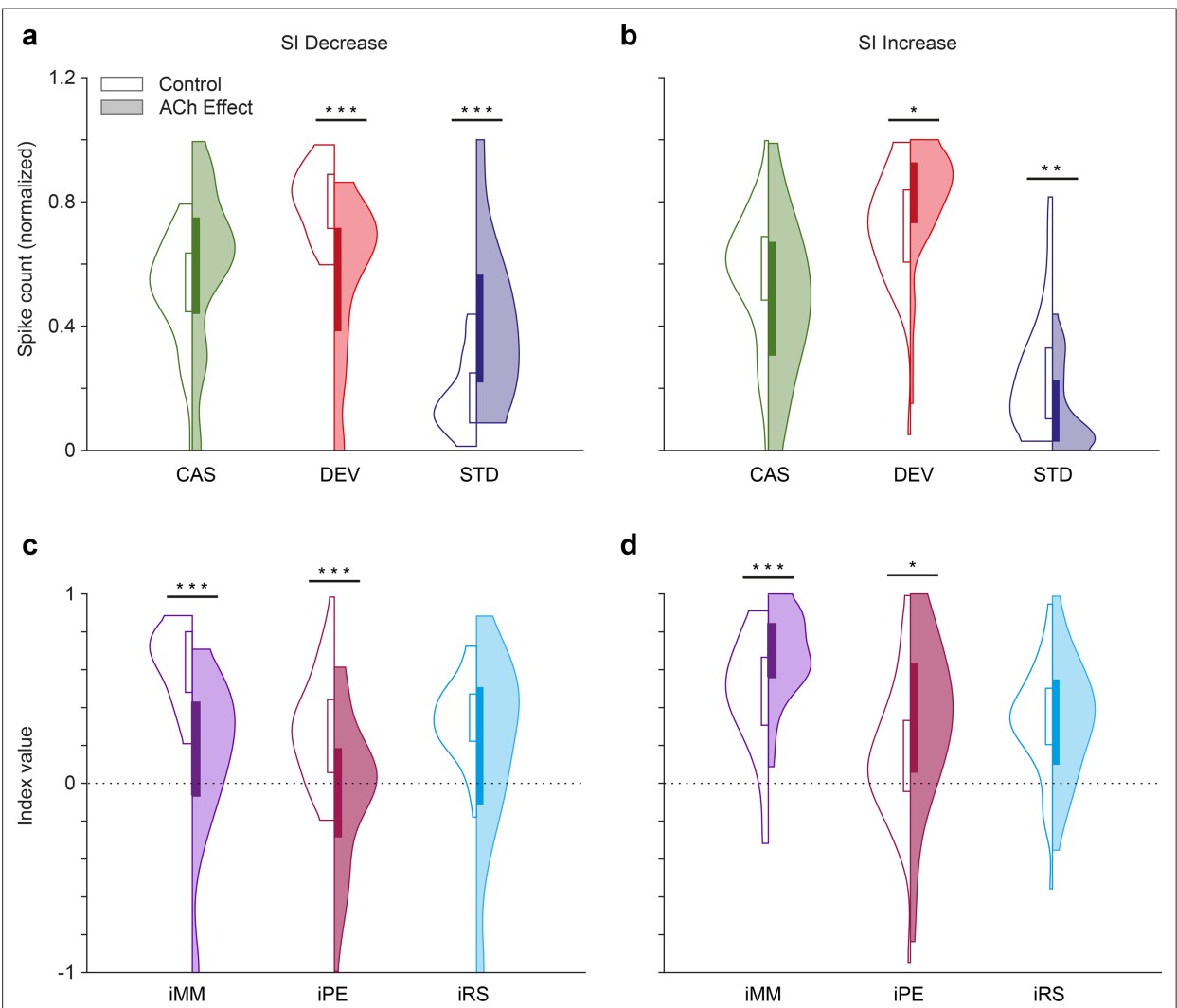

**Figure 7.** Predictive coding indices depending on the effect of acetylcholine (ACh) on selectivity index (SI). The application of ACh decreased the normalized spike counts in response to deviant stimuli (red) and increased the normalized spike counts in response to standard stimuli (blue) in those units where SI decreased due to the application of ACh (**a**), while the changes were opposite in those units where SI increased due to the application of ACh (**b**) In neither case, did the response to the cascade condition (green) change significantly, but there was a slight trend in the same direction as the changes in the standards. In consequence, the neuronal mismatch (iMM, purple) and prediction error indexes (iPE, light red) decreased significantly in those units where SI decreased due to the application of ACh (**c**), but increased in those units where SI increased due to the application of ACh (**d**) The repetition suppression index (iRS) in light blue, did not change significantly in any case. *p<0.05; **p<0.01; ***p<0.001.

The online version of this article includes the following source data and figure supplement(s) for figure 7:

**Source data 1.** Source data for *Figure 7*.

**Figure supplement 1.** Experimental paradigms and Interpretation of the control conditions.

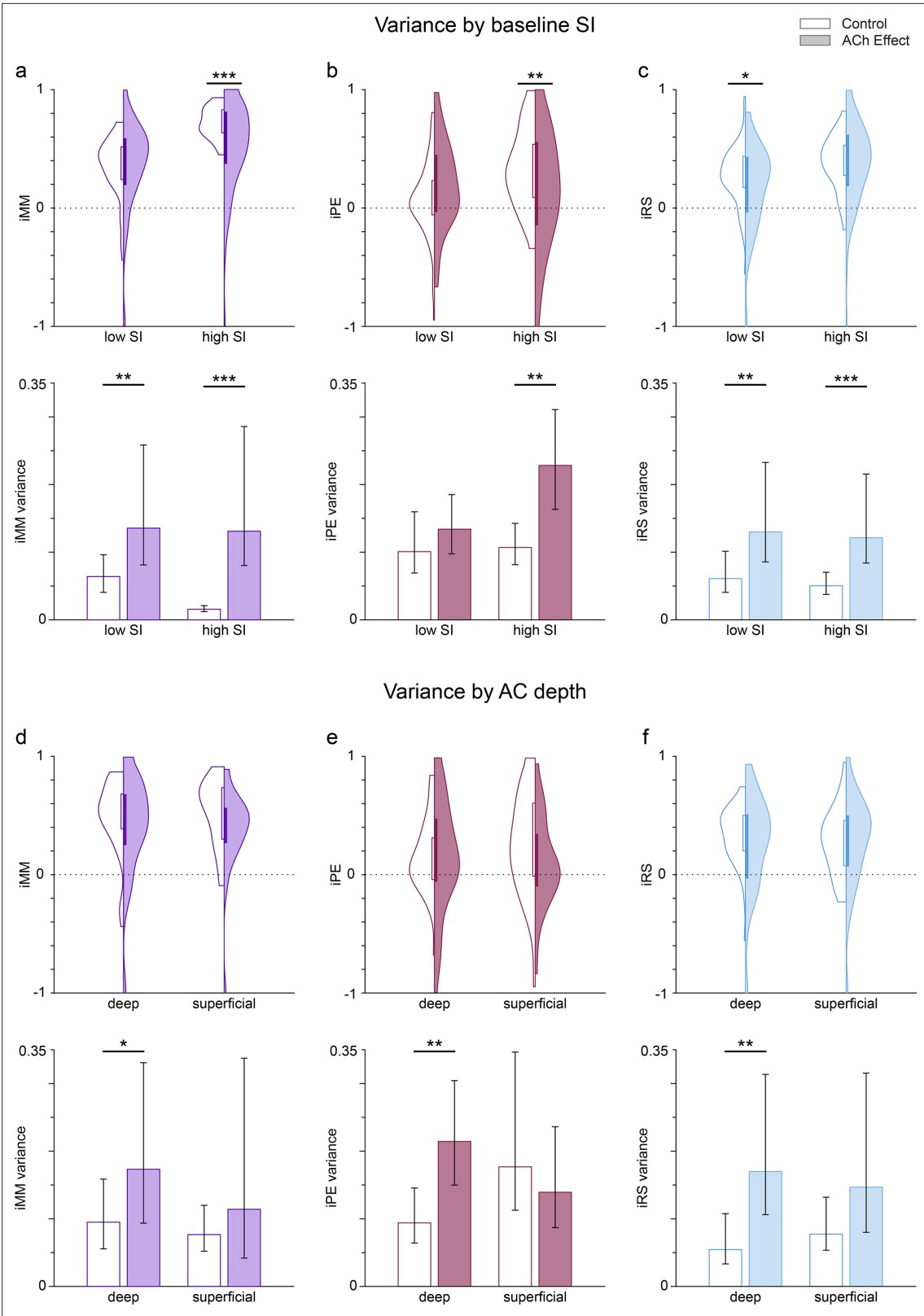

**Figure 8.** Dependence of the variance of predictive coding indexes on the selectivity index (SI) before acetylcholine (ACh) application, as well as and on auditory cortex (AC) depth. Violin plots show the distribution of the neuronal mismatch (iMM), prediction error (iPE), and repetition suppression (iRS) indexes based on the SI value before application (**a**-**c**; low SI: SI <0.6143; high SI: SI <0.6143, see text) or the AC depth (**d**-**f**; deep: >650 μm; superficial: <650 μm). Bar graphs show the variance of the related indexes. Error bars indicate 95% CI. The results before ACh application are shown on a white

*Figure 8 continued on next page*

*Figure 8 continued*

background (control), while shaded backgrounds represent the results during ACh application (effect). The application of ACh reduced significantly iMM and iPE (**a** and **b**, top row) for those frequencies with high baseline SI, as well as iRS (**c**, top row) for frequencies with low baseline SI. The variance of all indexes increased significantly during ACh application, except for the iPE of the low SI group. Regarding the depth of the recording sites, ACh did not significantly change the values of any indexes, neither for deep nor for superficial units (**d-f**, top row). However, ACh significantly increased the variance of all indexes for deep units, but not for superficial units (**d-f**, bottom row). *p<0.05; **p<0.01; ***p<0.001.

The online version of this article includes the following source data for figure 8:

**Source data 1.** Source data for *Figure 8*.

## ACh effect on nMM as a function of topographic distribution

A remaining question is whether the effect of ACh is uniformly distributed across AC neurons in different fields and layers. *Figure 9* shows the CSI (first row) and spike count levels for the standard and deviant tones (middle and bottom rows, respectively) before and after ACh application, as well as the difference between the ACh and control conditions. The highest levels of CSI in the control condition (left column) were found in SRAF. ACh application (middle column) produced an overall increment of CSI in AAF and a decrement in the rest of the fields, however, such changes did not reach statistical significance (p>0.79 in all fields, Wilcoxon signed-rank test with Holm-Bonferroni correction). Regarding the responses to the standard stimuli (middle row), ACh application caused an increase of the firing rate in A1 and PAF, and a decrease in AAF, VAF, and SRAF, but such changes

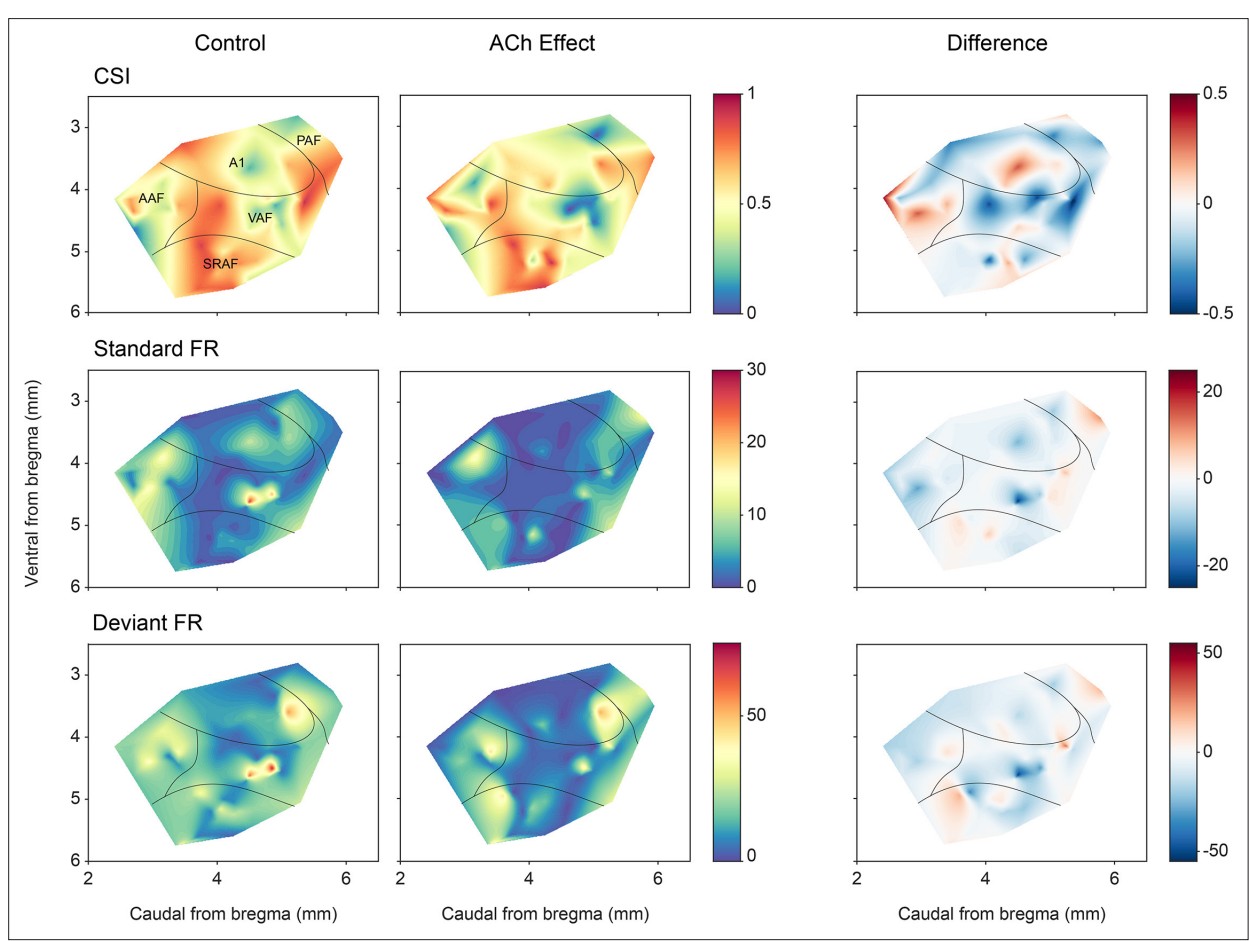

**Figure 9.** Anatomical localization of common SSA index (CSI) and responses to the deviant and standard tones. Distribution of CSI (top row) and responses (spikes per trial) for deviant (bottom row) and standard tones (middle row), before and after acetylcholine (ACh) application, as well as the difference between both conditions.

The online version of this article includes the following source data for figure 9:

**Source data 1.** Source data for *Figure 9*.

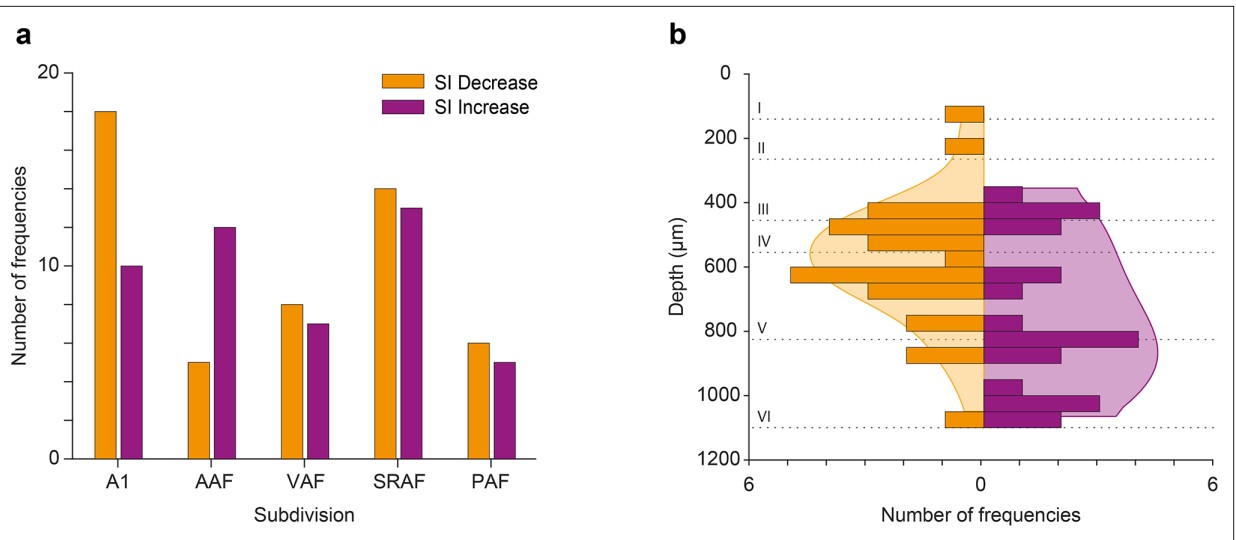

**Figure 10.** Effect of acetylcholine (ACh) on neuronal mismatch (nMM) in auditory cortex (AC) fields and layers. (**a**) Number of tested frequencies in which selectivity index (SI) decreased (orange) or increased (purple) for each AC field. In A1 it is more likely to find frequencies showing SI decrease, while in anterior auditory field (AAF) it is more likely to find those showing SI increase. In VAR, SRAF, and PAF, the likelihood of both types is similar. (**b**) Distribution of tested frequencies in which SI decreased (orange) or increased (purple) significantly due to the effect of ACh, across AC layers. The shaded background depicts the probability density functions. Note that units experiencing SI decrease are more prevalent around the granular layer (layer IV), while units experiencing SI increase are more commonly found in the infragranular layers.

The online version of this article includes the following source data for figure 10:

**Source data 1.** Source data for *Figure 10*.

were only significant in AAF (p=0.001, Wilcoxon signed-rank test with Holm-Bonferroni correction). By contrast, the firing rates in response to the deviant sounds (bottom row) decreased in all fields during ACh application, with significant changes in AAF, VAF, and SRAF (p=0.011, p=0.026, and p=0.009, respectively; Wilcoxon signed-rank test with Holm-Bonferroni correction).

In order to better understand the effect of ACh on the responses to the individual frequencies of the oddball paradigm ($f_1$ and $f_2$), we calculated the SSA index (SI) independently for each of the paradigm frequencies. We found that ACh was more likely to cause an SI decrease in all AC fields except for AAF, where ACh was more likely to cause an SI increase (*Figure 10a*; *Figure 10b*) shows the distribution of recording depths within the AC, for those units that showed a significant SI decrease (orange) or increase (purple) during the application of ACh. The shaded background shows the probability density function for those distributions. We found that frequencies that experienced an SI decrease during ACh application were most likely located in the granular layer (layer IV) or the upper infragranular regions (dorsal parts of layer V). In contrast, frequencies that experienced an SI decrease during ACh application were more likely located in deeper infragranular regions (layers V and VI).

## Discussion

We recorded neuronal responses from the AC under an auditory oddball paradigm while performing microiontophoretic applications of ACh to investigate the role of the cholinergic system in the processing of unexpected or surprising events. Furthermore, we employed cascade sequences to control for repetition suppression and assess cholinergic effects on prediction error relative to repetition suppression. Our results show that (1) ACh modulates nMM, evidenced by decrements or increments of the neuronal mismatch indexes (iMM, iPE, and iRS). The decreased SI is due to a diminished response to the deviant tone while increased SI is the consequence of a diminished response to the standard sounds (*Figure 6*, and examples in *Figures 2–3*); (2) ACh exerts a global and distributed effect on AC neurons that is layer-specific but field unspecific (*Figures 9 and 10*); (3) The decreased- and increased SI are driven by the effects of ACh on iPE but not iRS (*Figure 7*), and (4) ACh increases the variance of iMM, iPE, and iRS. Together, these findings reveal how ACh modulates prediction error

signals and repetition suppression differentially in the AC, likely gating these signals, as they ascend beyond the auditory cortex to frontal, higher cognitive regions.

ACh increases the amplitude of AC neuronal responses to sound stimulation, decreases the auditory threshold, and sharpens the receptive field in the AC. (*Edeline, 2012*; *Irvine, 2018a*; *Irvine, 2018b*; *Kilgard and Merzenich, 1998*; *Ma and Suga, 2005*; *Metherate, 2011*; *Metherate et al., 1992*; *Puckett et al., 2007*). These ACh-mediated effects are produced by a rapid disinhibition of neuronal responses, modifying synaptic strength, enhancing excitatory-inhibitory balance, and reorganizing cortical circuits promoting cortical plasticity (*Froemke et al., 2007*; *Irvine, 2018a*; *Irvine, 2018b*; *Metherate, 2011*; *Picciotto et al., 2012*). But ACh also produces direct inhibitory effects leading to a decrease of cortical responses (*Colangelo et al., 2019*; *Dasgupta et al., 2018*; *Eggermann and Feldmeyer, 2009*; *Qi and Feldmeyer, 2022*). Our results confirm these mixed effects (excitatory and inhibitory, during the application of ACh) at the neuronal level. However, when looking at the population level, the overall effect in our sample was a differential reduction of the responses to either deviant or standard stimuli. Apart from the direct hyperpolarizing effect of ACh on some cell types, it is also possible that some of the recorded neurons received direct inhibitory inputs, and those inhibitory neurons were the ones affected by ACh. By increasing the firing rates of such inhibitory interneurons, the observed overall effect of ACh could lead to a reduction of the activity of the recorded neurons. An important functional motif — for state modulation in the cortex — is through disinhibition consisting of parvalbumin-, somatostatin-, and vasoactive intestinal peptide- expressing interneurons. When the latter receives cholinergic projections from the basal forebrain, they remove the inhibition on layer 2/3 pyramidal neurons exerted by somatostatin interneurons (*Fu et al., 2014*). The connectivity relationships among inhibitory neurons seem to be critical. Moreover, it has been recently described how each of those cell types responds differently to deviants and standards in the auditory cortex (*Yarden et al., 2022*). ACh plays important roles in arousal, attention, and sensory learning (*Hasselmo, 1999*; *Hasselmo and Sarter, 2011*; *Metherate, 2011*; *Picciotto et al., 2012*; *Sarter et al., 2001*; *Weinberger, 2004*). Thus, it is not surprising that ACh may have a critical role in shaping nMM, which has many properties in common with behavioral habituation to a repeated stimulus and can be considered a simple form of learning (*Irvine, 2018a*; *Nelken, 2014*; *Netser et al., 2011*). Furthermore, a recent study in rats has shown that the administration of muscarinic ACh receptor agonists and antagonists affected MMN amplitude in A1 and PAF (*Schöbi et al., 2021*), suggesting that the cholinergic system modulates the connectivity between those areas during the occurrence of deviant sounds. But to the best of our knowledge, the present study demonstrates a specific effect of ACh on prediction error responses.

In the IC, ACh selectively increased the response to the repetitious standard stimulus (*Ayala and Malmierca, 2015*), while we show here that ACh in the AC not only increases the responses to the unexpected deviant sounds leading to an increase of SI (*Figure 6d*) but also decreases responses to the repeated standard that result in a diminished CSI index (*Figure 6c*). This divergent effect of ACh on subcortical and cortical nMM may be related to the different origin of ACh and/or the unique organization of neuronal circuitries in IC and AC (*Figure 11*). While the sources of ACh to IC emerge from the pedunculopontine and laterodorsal tegmental regions in the brainstem (*Motts and Schofield, 2009*), those to AC originate in the basal forebrain (*Chavez and Zaborszky, 2017*).

According to the predictive coding formulations, cortical areas send predictions to lower hierarchical areas to inhibit neuronal populations encoding prediction errors. When top-down predictions match bottom-up sensory input, the prediction error response is attenuated. Conversely, those lower areas send error information to higher hierarchical centers when event predictions fail to drive higher-level representations and improve the top-down predictions. If a stimulus is not predicted, the difference between predicted and received inputs yields large prediction errors, while if a stimulus is constantly repeated, sensory learning enables more accurate predictions and a decrease in prediction error responses – a phenomenon referred to as repetition suppression (*Auksztulewicz and Friston, 2016*). Thus, nMM can be explained under the predictive coding framework (*Carbajal and Malmierca, 2018*; *Carbajal and Malmierca, 2020*; *Parras et al., 2017*; *Malmierca et al., 2019*), and recent studies that found LFP and spiking responses to omitted tones in the rat auditory cortex (*Auksztulewicz et al., 2023*; *Lao-Rodríguez et al., 2023*), among others, strongly support this interpretation. However, predictions and prediction errors may also be modulated in a context-dependent manner, meaning that responses can be modulated and given precedence depending on the context in which

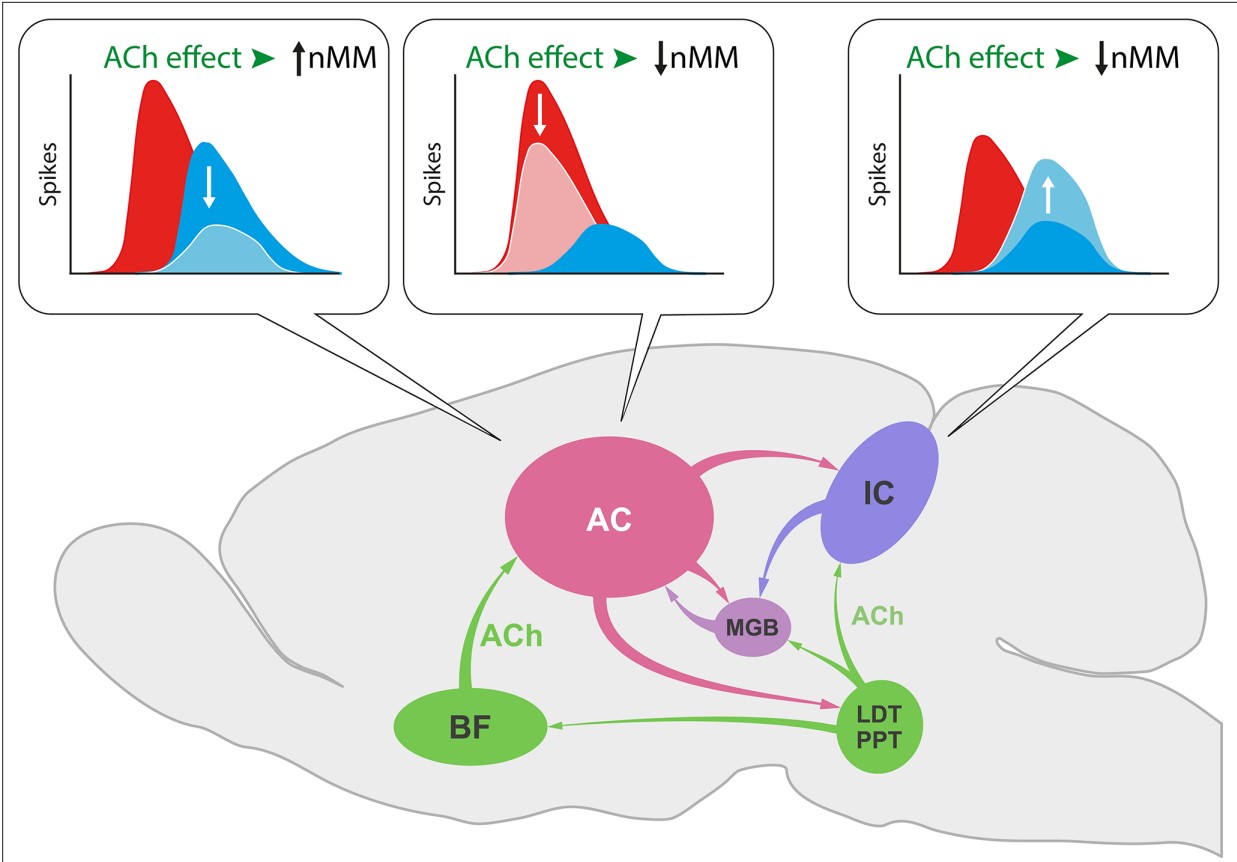

**Figure 11.** Effect of acetylcholine (ACh) on neuronal mismatch (nMM) in auditory midbrain and cortex. ACh increases nMM in the auditory cortex (AC) reducing the firing rate in response to either 'deviants' or 'standards,' while it shows a different effect in the IC, as ACh decreases nMM by increasing the responses to the 'standard' tones (*Ayala and Malmierca, 2015*). These divergent effects may be explained by the different connectivity of both auditory areas. While the AC receives its cholinergic input from the basal forebrain (BF), the IC is innervated by the pedunculopontine (PPT) and laterodorsal tegmental (LDT) regions in the brainstem. This intersection of auditory and cholinergic nuclei creates an intricate network involving ascending and descending projections that ultimately modulate the processing of auditory deviance detection.

the stimulus is perceived (*Keller and Mrsic-Flogel, 2018*). The source of such a modulating or gating signal may arise from neuromodulatory inputs (*Fu et al., 2014*; *Moran et al., 2013*; *Pinto et al., 2013*; *Polack et al., 2013*; *Thiele and Bellgrove, 2018*) that can not only gate plasticity (*Kilgard and Merzenich, 1998*; *Martins and Froemke, 2015*; *Weinberger, 2004*), but also change the balance of top-down versus bottom-up influence (*Yu and Dayan, 2002*).

Within the predictive coding framework, classical neuromodulators such as ACh are often thought to increase the precision of prediction error signaling (*Auksztulewicz et al., 2018*; *Auksztulewicz and Friston, 2016*; *Feldman and Friston, 2010*; *Moran et al., 2013*). Indeed, *Yu and Dayan, 2005*; *Yu and Dayan, 2002* proposed that ACh levels reflect the 'expected uncertainty' (when there is less confidence in the prediction) associated with top-down information and modulate the interaction between top-down and bottom-up processing in determining the appropriate neural representations for inputs (*Auksztulewicz et al., 2017*; *Yu and Dayan, 2005*; *Yu and Dayan, 2002*). Accordingly, the precision of prediction errors would encode the confidence or reliability afforded by such errors (*Friston, 2005*), increasing the influence of precise prediction errors on higher-level processing. Thus, the canonical view – within the predictive coding framework — is that precision would affect postsynaptic gain through the action of neuromodulators such as ACh, enhancing the responses of units broadcasting prediction errors. Using computational neuronal modeling and EEG recordings under oddball paradigms in subjects manipulated pharmacologically with galantamine a competitive inhibitor of acetylcholinesterase, *Moran et al., 2013* have shown that ACh enhances the precision of bottom-up synaptic transmission in cortical hierarchies by optimizing the gain of supragranular

pyramidal neurons, amounting to an increase in the precision of prediction error signaling. Thus, our results and those from Moran and colleagues (2013) suggest that ACh mediates the representation of precision.

Our results are consistent with this predictive coding picture; particularly when noting that an increase in precision leads to a representational sharpening (*Kok et al., 2012a*), in which the majority of (prediction error) neurons are suppressed and a few neurons encoding precise prediction errors are enhanced (*Friston, 2018*; *Kok et al., 2012a*; *Shipp, 2016*). This is consistent with the effect of acetylcholine on average over neurons studied and the differential effect on standard and oddball responses; dependent upon their SI.

ACh increases the variance of the prediction error component of neuronal responses in those units with higher SI (*Figure 8b*, top), especially in units with strong deviance detection properties (*Figure 8b*, bottom). This finding suggests that ACh plays a key role in encoding precision or uncertainty. Interestingly, ACh not only increases the variance of prediction error responses, but also the variance of repetition suppression (*Figure 8c*).

A change in the precision of prediction errors may denote — or alert to — a variable environment, where it is difficult to extract rules and only generate weak (and less useful) predictions. Under this scenario, some fluctuations in the sensory input can be expected, assigning less precision to prediction errors (*Yon and Frith, 2021*). If so, ACh would adjust the predictions' confidence, so we would not be surprised by small changes in the environment, since they are likely, expected noisy, fluctuations. Otherwise, an excessive confidence on predictions in a volatile environment may override the incoming evidence, leading to the false inference of expected by absent events (*Yon and Frith, 2021*). Attention is thought to boost the precision of prediction errors (*Kok et al., 2012b*), and our experiments carried out under anesthesia clearly lack attention, where baseline cholinergic activity is low, making it difficult to interpret how the results generalize to more ecologically valid situations (*Minces et al., 2017*). Interestingly, there is a wealth of data suggesting that minimal impairments to the encoding or defective signaling of precision or uncertainty are sufficient to explain psychopathic conditions such as schizophrenia or autism (*Friston, 2023*). Furthermore, while we are not aware of any known direct connection between auditory processing disorders and ACh, individuals with auditory processing disorders do have difficulties with auditory selective attention, so perhaps one could speculate that ACh, by modulating SSA/prediction error, could impact encoding of salient events, and if disrupted could lead to problems with selective attention. *Moore, 2006*; *Moore, 2012* speculated that auditory processing disorders may arise from unbalanced processing in bottom-up and top-down contributions, i.e., there might be some aspects of predictive processing altered. A recent study (*Felix et al., 2019*) analyzed altered temporal processing at the level of the brainstem in $\alpha_7$-subunit of the nicotinic acetylcholine receptor ($\alpha_7$-nAChR)-deficient mice. After studying $\alpha_7$-nAChR knockout mice of both sexes and wild-type colony controls, they concluded that the malfunction of the *CHRNA7* gene that encodes the $\alpha_7$-nAChR may contribute to degraded spike timing in the midbrain, which may underlie the observed timing delay in the ABR signals. These authors (*Felix et al., 2019*) proposed that their findings are consistent with a role for the $\alpha_7$-nAChR in types of neurodevelopmental and auditory processing disorders. There is also evidence of cholinergic system disfunction being related to the pathophysiology of Alzheimer's disease (*Pérez-González et al., 2022*). For instance, disfunction of the synapses of cholinergic neurons in the hippocampus and nucleus basalis of Meynert, as well as decreased choline acetyltransferase activity, is associated with memory disorders in Alzheimer's disease (*Hampel et al., 2018*). Also, Alzheimer's disease patients show reduced amounts of the vesicular ACh transporter in some brain areas (*Aghourian et al., 2017*). Finally, cholinesterase inhibitors seem to have some favorable effects in the treatment of Alzheimer's disease patients (*Sharma, 2019*).

It is unclear to what degree the amount of agonist liberated used iontophoretic application of cholinergic agonists emulates a relevant physiological condition (*Minces et al., 2017*). The application of exogenous ACh, as we used here, could potentially surpass in physiological concentrations, making the results hard to extrapolate to a behaving animal. Furthermore, future studies should investigate the effects of different doses of ACh on auditory processing to determine if there is a dose-response relationship as well as the effect of muscarinic- and nicotinic receptors blockade to theoretically observe more akin, analogous physiological effects. Similarly, forthcoming studies using optogenetic techniques to activate only basal forebrain cholinergic neurons while also recording from auditory cortex neurons will allow a fine-grained investigation of the role of cortical acetylcholine on sensory

representation. Moreover, because our study was carried out in anesthetized animals, it precludes generalization of the effects of ACh on behaving states (**Quintela-Vega et al., 2023**). Future studies using an awake preparation will confirm a definitive corroboration of the effect of ACh on precision and determine the behavioral relevance of these findings. The available neurobiological evidence suggests that precision is encoded by the synaptic gain of superficial pyramidal cells encoding prediction errors, which are controlled by neuromodulatory systems (e.g. dopaminergic, cholinergic) and/or synchronized neural activity (**Feldman and Friston, 2010**; **Kok et al., 2012b**). However, we do not find changes in prediction error precision in the superficial layers; our analysis suggests instead that the effects of ACh were more limited to the deep AC layers (**Figure 8e**). We observed a distinct and differential distribution of units throughout the cortical layers, contingent upon the influence of ACh on their selectivity index (SI). Units exhibiting an SI decrease during ACh application were predominantly located and biased in layers III-IV and its adjacent regions, while those demonstrating an SI increment during ACh application spanned from the deeper layer IV to layer VI (**Figure 10b**). In the initial conceptualizations of predictive coding, it was posited that prediction units reside in the infragranular layers (layers V and VI), with error units situated in supragranular layers (layers I – III) (**Bastos et al., 2012**; **Friston, 2005**). Alternatively, some models suggest that error units would be localized in the granular layer (layer IV), while prediction units would be detected in both supra- and infragranular layers biased competition model of predictive coding (**Spratling, 2008**). In our recordings, units displaying an SI reduction during ACh application were also those with a higher SI under baseline conditions (i.e. larger response to deviant than to standard stimuli), thereby potentially qualifying as error units. These units were identified in layer IV and neighboring areas, which appears to support the model proposed by **Spratling, 2008**. Furthermore, according to **Eggermann and Feldmeyer, 2009**, the application of ACh would filter weak sensory inputs to layer IV and amplify signals at superficial and deep layers, effectively increasing the signal-to-noise ratio. In contrast, units demonstrating an SI increase during ACh application could be considered potential prediction units; such units were detected in granular and deep layers but were absent in superficial layers. This observation aligns with both predictive coding models, as each postulates that deep layers contain prediction units. A very interesting study would have been to estimate whether the neurons we recorded were excitatory or inhibitory based on the spike shape. Indeed, we have previously shown that both types of responses show prediction error (**Pérez-González et al., 2021**) but ACh may affect them differentially and this potential divergence may explain some of the variance of the data. This would involve implementing a similar analysis as done in our previous study (**Pérez-González et al., 2021**), however, that study used single-unit recordings and here we used multi-unit recordings. In the present data, the recordings captured occasional spikes from other neurons which introduced alterations in the average spike shape, and thus precluded an accurate categorization. Therefore, we decided to discard this analysis for the sake of accuracy and appropriate scrutiny awaits future studies.

In conclusion, we have shown that ACh plays a multifold role in modulating nMM in AC, affecting the precision of prediction error signaling and gating prediction errors to hierarchically higher processing levels.

# Materials and methods
## Subjects and surgical procedures

The experimental protocols were approved conforming to the University of Salamanca Animal Care Committee standards and the European Union (Directive 2010/63/EU) for the use of animals in neuroscience research. Experiments were performed on 37 adult female Long-Evans rats with body weights within 180–250 g. We used female rats as our previous work is based on this sex but no attempt to study gender dimension was aimed, which would have required a larger number of animals. Surgical anesthesia was induced and maintained with urethane (1.5 g/kg, intraperitoneal), with supplementary doses (0.5 g/kg, intraperitoneal) given as needed. Dexamethasone (0.25 mg/kg) and atropine sulfate (0.1 mg/kg) were administered at the beginning of the surgery to reduce brain edema and the viscosity of bronchial secretions, respectively. Normal hearing was verified with auditory brainstem responses (ABR) recorded with subcutaneous needle electrodes, using a RZ6 Multi I/O Processor (Tucker-Davis Technologies, TDT) and processed with BioSig software (TDT). ABR stimuli consisted of 100 µs clicks at a 21 /s rate, delivered monaurally to the right ear in 10 dB steps, from 10 to 90 decibels

of sound pressure level (dB SPL), using a closed-field speaker. After the animal reached a surgical plane of anesthesia, the trachea was cannulated for artificial ventilation and a cisternal drain was introduced to prevent brain hernia and brain edema. Isotonic glucosaline solution was administered periodically (5–10 ml every 7 hr, subcutaneous) throughout the experiment to prevent dehydration. Body temperature was monitored with a rectal probe thermometer and maintained between 37 and 38°C with a homeothermic blanket system.

The auditory cortex surgery was described in previous works (*Nieto-Diego and Malmierca, 2016*; *Parras et al., 2017*). The temporal bone was exposed and the auditory cortex was located using stereotactic coordinates (*Murone et al., 1997*). A craniotomy was performed over the auditory cortex, the dura was removed carefully, and the exposed area was filled with a layer of agar to prevent desiccation and to stabilize the recordings. Before applying the agar, a magnified picture (25x) of the exposed cortex was taken with a digital camera coupled to the surgical microscope (Zeiss) through a lens adapter (TTI Medical). The picture included a pair of reference points previously marked on the dorsal ridge of the temporal bone, indicating the absolute scale and position of the image with respect to bregma (the reference point). This picture was displayed on a computer screen and was overlapped with a micrometric grid, over which the placement of the multibarrel for every recording was marked. The micrometric grid allowed the generation of coordinates in a two-dimensional axis, from the super-imposed image of the auditory cortex of each animal. Once the coordinates of each of the recorded units of all the animals were obtained, we used the functions 'griddata' and 'contourf' of MATLAB to generate topographic maps (through a graduation of colors) of the characteristic frequency (CF), CSI levels, and deviant and standard tone responses in all auditory cortical fields.

## Electrophysiological recording and microiontophoresis

A tungsten electrode (1–3 MΩ) was used to record multiunit neuronal activity. It was attached to a five barrel multibarrel borosilicate glass pipette that carried the drug solution to be delivered in the vicinity of the recorded neuron. The multibarrel's tip was cut to a diameter of 20–30 µm. The center barrel was filled with saline solution for current compensation (165 mM NaCl) whereas the other barrels were filled with 1 M acetylcholine chloride (Sigma, catalog no. A6625). The pH was adjusted between 4.0–4.2 (*Ayala and Malmierca, 2015*). Drugs were retained by applying a –15 nA current, and were ejected when required, typically using 30–40 nA currents (Neurophore BH-2 System, Harvard Apparatus). The duration of the drug ejection usually lasted 8–10 min and the recording protocols were extended until the effect of the drug had disappeared (60–90 min approx.). The multibarrel assembly was positioned over the pial surface of the auditory cortex, forming a 30° angle with the horizontal plane towards the rostral direction, and advanced using a piezoelectric micromanipulator (Sensapex) until we observed a strong spiking activity synchronized with the train of search stimuli made of white noise. Analog signals were digitalized with a RZ6 Multi I/O Processor, a RA16PA Medusa Preamplifier, and a ZC16 headstage (TDT) at a 12 kHz sampling rate and amplified 251 x. Neurophysiological signals for multiunit activity were band-pass filtered between 0.5 and 4.5 kHz.

## Experimental design and stimulation paradigms

Sound stimuli were generated using the RZ6 Multi I/O Processor (TDT) and custom software programmed with the OpenEx Suite (TDT) and MATLAB. Sounds were presented monaurally through a speaker, in a closed-field condition to the ear contralateral to the left auditory cortex. We calibrated the speaker to ensure a flat spectrum up to ~75 dB SPL between 0.5 and 44 kHz; the second and third harmonics were at least 40 dB lower than the fundamental at the loudest output level for all the frequencies. The experimental stimuli were pure tones in the range 0.5–44 kHz, with a duration of 75 ms, including 5 ms rise/fall ramps presented at a rate of 4 stimuli/s. Once a suitable neuron was found, the frequency response area (FRA) of the cell, i.e., the combination of frequencies and intensities capable of evoking a suprathreshold response, was obtained automatically using a randomized paradigm that presented tones between 0.5–44 kHz in 25 logarithmic steps, with intensities spaced by 10 dB steps, from 0 to 70 dB SPL, at a repetition rate of 4 /s. Based on this information, we selected a pair of frequencies evoking similar responses at 10–30 dB above the threshold. We used pure tones at these frequencies as the stimuli in the oddball paradigm. Oddball sequences consisted of frequently repeating stimuli (standard tones) which were pseudo-randomly interleaved with rare events (deviant tones). Two oddball sequences with fixed parameters (400 trials each, 75 ms stimulus duration, 0.5

octaves frequency separation, 10% deviant probability, 250 ms onset to onset, and a minimum of three standard tones before a deviant) were presented for every pair of stimuli thus selected. In one of the sequences, the low frequency ($f_1$) was the 'standard' and the high frequency ($f_2$) was the 'deviant,' and in the other sequence, their roles were inverted. The order of presentation of these two sequences was randomized across sites. In some cases, one or more extra pairs of stimuli were selected, and the oddball sequences were repeated with the new stimuli.

According to the predictive coding framework, mismatch responses like those obtained during an oddball paradigm can be divided into two components: repetition suppression, a reduction in the response caused by a repeated stimulus, and prediction error, an increased response caused by the violation of a regularity (*Carbajal and Malmierca, 2018*). In a subset of the experiments, we used control sequences to evaluate the separate contribution of repetition suppression and prediction error. Control sequences consisted of 10 tones evenly spaced by 0.5 octaves (same as in the oddball sequences), including the tones used in the oddball paradigm, and all stimuli at the same previously chosen sound level. Each control sequence lasted 400 trials, the duration of all stimuli was 75 ms, and the presentation rate 4 /s. We used two different control sequences, namely the many-standards and cascade sequences (*Figure 7—figure supplement 1*). The many-standards control is the consecutive presentation of blocks of 10 tones randomly ordered within the block, each tone with a 10% probability of occurrence (*Schröger and Wolff, 1996*). In this sequence the tones are unpredictable, and it is not possible to establish a rule which could be used to predict the following tones. On the other hand, the cascade control consists of the regular presentation of the same 10 tones in ascending or descending frequency succession (*Ruhnau et al., 2012*). This sequence also avoids the effects of the repetition of a single stimulus, but in this case, it maintains a predictable context. Cascade sequences are considered a more rigorous control than many-standard sequences because, like the oddball sequence, a regularity is established (*Carbajal and Malmierca, 2018*).

## Data analysis

We calculated the bandwidth of the FRAs (*Figure 4*) for each recorded unit by measuring the width of the response area in octaves (relative to the low-frequency border), at 10 and 30 dB above the minimum threshold (BW10 and BW30, respectively).

The degree of nMM was quantified by the CSI, reported previously (*Ayala et al., 2015*; *Duque et al., 2016*; *Malmierca et al., 2009*; *Ulanovsky et al., 2004*; *Ulanovsky et al., 2003*). The CSI reflects the difference between the neural responses to the deviant and standard stimuli, normalized to the total of responses to both stimuli, and is defined as:

$$CSI = \frac{d(f1) + d(f2) - s(f1) - s(f2)}{d(f1) + d(f2) + s(f1) + s(f2)}$$

where $d(f_i)$ and $s(f_i)$ are responses to each frequency $f_1$ or $f_2$ when they were the deviant *(d)* or standard *(s)* stimulus in the oddball paradigm, respectively.

Similarly, a frequency-specific SSA index (SI) was calculated for each of the frequencies presented in an oddball paradigm, as:

$$SI = \frac{d(f) - s(f)}{d(f) + s(f)}$$

Where *d(f)* and *s(f)* are responses to frequency *f* when it was the deviant *(d)* or standard *(s)* stimulus in the oddball paradigm, respectively.

For the subset of experiments where we recorded the many-standards and cascade controls, we compared the responses to the same physical stimulus when it took the role of a standard, a deviant (ascending or descending, depending on whether the preceding standard was of lower or higher frequency, respectively), or as part of a cascade sequence (matching the ascending or descending condition of the corresponding deviant). Alongside the oddball paradigm, we recorded responses of neurons to two cascade sequences (ascending and descending), which consisted of 10 tones selected within the FRA presented in a predictable succession of increasing or decreasing frequencies. We did not include the many-standards control in these analyses because these experiments are time-consuming and the need of holding the recording neuron for long enough before, during and after the drug injection. However, we have previously demonstrated that the many-standards and cascade

control responses were comparable, and the latter is considered to be more rigorous. (*Casado-Román et al., 2019*; *Parras et al., 2017*; *Ruhnau et al., 2012*; *Valdés-Baizabal et al., 2019*).

The responses of each neuron were normalized for each tested frequency as

$$Normalized\ Deviant = Deviant/N$$
$$Normalized\ Standard = Standard/N$$
$$Normalized\ Control = Control/N$$

where,

$$N = \sqrt{Deviant^2 + Standard^2 + Control^2}$$

is the Euclidean norm of the vector defined by the deviant, standard, and cascade responses, so that the normalized responses take values in the range of 0–1.

With these normalized baseline-corrected spike counts, we next computed the indices of neuronal mismatch (iMM), repetition suppression (iRS), and prediction error (iPE) as:

$$iMM = Normalized\ Deviant - Normalized\ Standard$$
$$iRS = Normalized\ Control - Normalized\ Standard$$
$$iPE = Normalized\ Deviant - Normalized\ Control$$

Index values ranged between –1 and 1 and facilitated the quantitative decomposition of neuronal mismatch into repetition suppression and prediction error since

$$iMM = iRS + iPE$$

Which is largely comparable to the CSI calculated for the oddball paradigm (*Parras et al., 2017*). In order to determine a significant effect of the drugs on the CSI (*Figure 5a*), we calculated an empirical distribution of CSI values by performing 2000 bootstraps of the responses to each standard or deviant stimulus for each neuron in the control condition, from which we determined a 95% confidence interval (vertical lines in *Figure 5a*). An effect was considered significant at $\alpha$=0.05 if the CSI during the drug condition fell beyond the limits of the control CSI confidence interval (red dots in *Figure 5a*). An equivalent approach was followed for the SI values in *Figure 6a*.

The area under the ROC curve (AUC) was calculated using the firing rates in response to deviant or standard stimuli as predictors (*Figures 5d, 6c and d*). Mutual Information on the firing rates in response to deviant or standard stimuli was calculated using the Information Breakdown Toolbox (*Magri et al., 2009*).

The confidence intervals for the variance of the different index distributions (*Figure 8*), were obtained by performing 1000 bootstraps.

Since parameters did not follow a normal distribution, non-parametric statistics were used: Mann-Whitney (for independent data) or Wilcoxon signed-rank test (for paired data). Multiple comparisons were corrected using the Holm-Bonferroni approach. Unless stated otherwise, all data are reported as mean ± SD. All of the data analyses were performed with Sigma Plot v12.5 and MATLAB software, using the built-in functions, the Statistics and Machine Learning toolbox, or custom scripts and functions developed in our laboratory. Custom software used in this paper is available upon reasonable request to the authors.

## Acknowledgements

We thank Drs. Francisco Garcia-Rosales and Julio Hechavarria for their advice in the mutual information analysis and Drs. Ryszard Auksztulewicz, Yaneri A Ayala, Donald Caspary, Diego Contreras, Karl Friston, Lauren Harms, and Marina Picciotto for their constructive criticisms of a previous version.

## Additional information

### Funding

| Funder | Grant reference number | Author |
|---|---|---|
| Agencia Estatal de Investigación | PID2019-104570RB-I00 | Manuel S Malmierca Ana Belén Lao-Rodríguez |
| Fundación Ramón Areces | CIVP20A6616 | Manuel S Malmierca David Pérez-González |
| Horizon 2020 Framework Programme | No 952378-BrainTwin | Manuel S Malmierca Ana Belén Lao-Rodríguez David Pérez-González |

The funders had no role in study design, data collection and interpretation, or the decision to submit the work for publication.

### Author contributions
David Pérez-González, Conceptualization, Data curation, Software, Formal analysis, Validation, Visualization, Methodology, Writing – original draft, Writing – review and editing; Ana Belén Lao-Rodríguez, Conceptualization, Data curation, Formal analysis, Investigation, Visualization, Methodology, Writing – review and editing; Cristian Aedo-Sánchez, Investigation, Writing – original draft, Writing – review and editing; Manuel S Malmierca, Conceptualization, Supervision, Funding acquisition, Visualization, Methodology, Writing – original draft, Project administration, Writing – review and editing

### Author ORCIDs
David Pérez-González (iD) http://orcid.org/0000-0001-6288-2692
Ana Belén Lao-Rodríguez (iD) http://orcid.org/0000-0002-4093-0160
Cristian Aedo-Sánchez (iD) http://orcid.org/0000-0003-0670-7843
Manuel S Malmierca (iD) https://orcid.org/0000-0003-0168-7572

### Ethics
All experimental procedures were carried out at the University of Salamanca and all methodological procedures were approved by the Bioethics Committee for Animal Care of the University of Salamanca (USAL-ID-574 and 683) and performed in compliance with the standards of the European Convention ETS 123, the European Union Directive 2010/63/EU, and the Spanish Royal Decree 53/2013 for the use of animals in scientific research.

Reviewer #1 (Public Review): https://doi.org/10.7554/eLife.91475.3.sa1
Reviewer #2 (Public Review): https://doi.org/10.7554/eLife.91475.3.sa2
Author Response https://doi.org/10.7554/eLife.91475.3.sa3

## Additional files

### Supplementary files
• MDAR checklist

### Data availability
Source data in the manuscript contains the numerical data used to generate the figures.

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
