## [Editor Report · eLife assessment]

The findings of this study are **valuable** as they provide new insights into the role of acetylcholine in modulating sensory processing in the auditory cortex. This paper reports a systematic measurement of cell activity in the auditory cortex before and after the microiontophoretic application of Ach during an oddball and cascade sequence of auditory stimuli. The evidence presented is **convincing**, as the study used a rigorous experimental design and statistical analysis. The manuscript will interest researchers in auditory neuroscience and neuromodulation, as well as clinicians and individuals with auditory processing disorders.

---

## [Referee Report · Reviewer #1 (Public Review)]

Summary:

This study examined the impact of exogenous microapplication of acetylcholine (Ach) on metrics of novelty detection in the anesthetized rat auditory cortex. The authors found that the majority of units showed some degree of modulation of novelty detection, with roughly similar numbers showing enhanced novelty detection, suppressed novelty detection or no change. Enhanced novelty responses were driven by increases in repetition suppression. Suppressed novelty responses were driven by deviance suppression. There were no compelling differences seen between auditory cortical subfields or layers, though there was heterogeneity in the Ach effects within subfields. Overall, these findings are important because they suggest that fluctations in cortical Ach, which are known to occur during changes in arousal or attentional states, will likely influence the capacity for individual auditory cortical neurons to respond to novel stimuli.

Strengths:

The work addresses an important problem in auditory neuroscience. The main strengths of the study are that the work appears to be systematically done with appropriate controls (cascaded stimuli) and utilizes a classical approach that ensures that drug application is isolated to the micro-environment of the recorded neuron. In addition, the authors do not isolate their study to only primary auditory cortex, but examine the impact of Ach across all known auditory cortical subfields.

Weaknesses:

1. As acknowledged by the authors, this study explicitly examines a phenomenon of high relevance to active listening, but is done in anesthetized animals, limiting its applicability to the waking state.

2. The authors do not make any attempt to determine, by spike shape/duration, if their units are excitatory or inhibitory, which may explain some of the variance of the data.

3. The application of exogenous Ach, potentially in supra-physiological amounts, makes this study hard to extrapolate to a behaving animal. A more compelling design would be to block Ach, particularly at particular receptor types, to determine the effect of endogenous Ach

---

## [Referee Report · Reviewer #2 (Public Review)]

Summary:

In this study, the authors investigate the effect of ACh on neuronal responses in the auditory cortex of anesthetized rats during an auditory oddball task. The paradigm consisted of two pure tones (selected from the frequency responses at each recording site) presented in a pseudo-random sequence. One tone was presented frequently (the "standard" tone) and the other infrequently (the "deviant" tone). The authors found that ACh enhances the detection of unexpected stimuli in the auditory environment by increasing or decreasing the neuronal responses to deviant and standard tones.

Strengths:

The study includes the use of appropriate and validated methodology in line with the current state-of-the-art, rigorous statistical analysis and the demonstration of the effects of acetylcholine on auditory processing.

Weaknesses:

The study was conducted in anesthetized rats, and further research is needed to determine the behavioral relevance of these findings.

---

## [Author Response]

The following is the authors’ response to the original reviews.

**eLife assessment**
The findings of this study are valuable as they provide new insights into the role of acetylcholine in modulating sensory processing in the auditory cortex. This paper reports a systematic measurement of cell activity in the auditory cortex before and after applying ACh during an oddball and cascade sequence of auditory stimuli in anesthetized rats. The results presented are solid given the rigorous experimental design and statistical analysis. The conclusions are provocative and will interest researchers in auditory neuroscience and neuromodulation, as well as clinicians and individuals with auditory processing disorders. However, the findings support multiple interpretations, beyond that offered by the authors.

Our reply: First and foremost, we would like to thank the editors and reviewers for their constructive criticisms, as well as their thoughtful and thorough evaluations of our manuscript. We greatly appreciate their assessment about the novelty and general significance in our study and have revised the manuscript according to their recommendations. In the following we include detailed responses and revisions based on the reviewer’s recommendations.

**Public Reviews:**

**Reviewer #1 (Public Review):**
Summary:This study examined the impact of exogenous microapplication of acetylcholine (Ach) on metrics of novelty detection in the anesthetized rat auditory cortex. The authors found that the majority of units showed some degree of modulation of novelty detection, with roughly similar numbers showing enhanced novelty detection, suppressed novelty detection, or no change. Enhanced novelty responses were driven by increases in repetition suppression. Suppressed novelty responses were driven by deviance suppression. There were no compelling differences seen between auditory cortical subfields or layers, though there was heterogeneity in the Ach effects within subfields. Overall, these findings are important because they suggest that fluctuations in cortical Ach, which are known to occur during changes in arousal or attentional states, will likely influence the capacity of individual auditory cortical neurons to respond to novel stimuli.Strengths:The work addresses an important problem in auditory neuroscience. The main strengths of the study are that the work was systematically done with appropriate controls (cascaded stimuli) and utilizes a classical approach that ensures that drug application is isolated to the micro-environment of the recorded neuron. In addition, the authors do not isolate their study to only the primary auditory cortex, but examine the impact of Ach across all known auditory cortical subfields.

Our reply: Thank you very much for these supportive comments and the appreciation of our work.

Weaknesses:1. As acknowledged by the authors, this study explicitly examines a phenomenon of high relevance to active listening but is done in anesthetized animals, limiting its applicability to the waking state.

Our reply: We agree; and indeed, this weakness was already recognized in the original manuscript but is now emphasized in the discussion.

1. The authors do not make any attempt to determine, by spike shape/duration, if their units are excitatory or inhibitory, which may explain some of the variance of the data.

Our reply: This is a very interesting question, and in fact, we have previously estimated whether neurons are excitatory or inhibitory based on the spike shape (Pérez-Gonzalez et al., 2021). Originally, we sought to implement a similar analysis here and tried to estimate if the recorded units were excitatory or inhibitory based on the spike shapes. But when we tried to perform this analysis, we found that in many cases the recordings had captured occasional spikes from other neurons. This caveat had introduced alterations in the average spike shape, and thus precluded an accurate categorization. Therefore, we decided to discard this analysis for the sake of correctness. This weakness is further commented on in the discussion.

1. The application of exogenous Ach, potentially in supra-physiological amounts, makes this study hard to extrapolate to a behaving animal. A more compelling design would be to block Ach, particularly at particular receptor types, to determine the effect of endogenous Ach.

Our reply: We agree again with the reviewer; this weakness was already acknowledged, but this is now further highlighted in discussion where we comment that future studies should analyze the effect of muscarinic- and nicotinic- receptors and blockade them to potentially observe more physiologically-comparable effects. Moreover, this issue is also related to a comment raised by reviewer#2 on a possible ‘dose-response relationship’ issue.

**Reviewer #2 (Public Review):**
Summary:In this study, the authors investigate the effect of ACh on neuronal responses in the auditory cortex of anesthetized rats during an auditory oddball task. The paradigm consisted of two pure tones (selected from the frequency responses at each recording site) presented in a pseudo-random sequence. One tone was presented frequently (the "standard" tone) and the other infrequently (the "deviant" tone). The authors found that ACh enhances the detection of unexpected stimuli in the auditory environment by increasing or decreasing the neuronal responses to deviant and standard tones.Strengths:The study includes the use of appropriate and validated methodology in line with the current state-of-the-art, rigorous statistical analysis, and the demonstration of the effects of acetylcholine on auditory processing.

Our reply: Thank you very much for these supportive comments and the appreciation of our work.

Weaknesses:The study was conducted in anesthetized rats, and further research is needed to determine the behavioral relevance of these findings.

Our reply: We agree; and indeed, this weakness was already recognized but is now emphasized in discussion.

**Reviewer #1 (Recommendations For The Authors):**
As outlined above, breaking out the units into those that are putative excitatory or inhibitory cells would be helpful, if possible. Other critiques are minor:1. "Acetylcholine", "ACh" and "Ach" are used throughout the manuscript. Please define the chosen abbreviation at first use, and be consistent.1. Line 116, remove comma after "ACh".1. Line 123, I would add "in the rat at the end of the first sentence since the species was not mentioned up to this point.1. Fig 2 - it would be useful in the Figure (not just in the text) to label red as being the deviant tone and blue as being the standard.1. In many Figures (e.g., Fig 5), the term "effect" is found in the legend rather than "ACh". It would seem more intuitive to label these as "ACh".1. The AUC and MI interpretations are not clear. Both are metrics that quantify similarity but the authors state that when these values decrease the neurons are less able to discriminate between them (i.e., they are more similar). Some clarifying text would be useful.1. L276 - should "SI increase" be "SI decrease"?1. L285 - would replace "solely" with "primarily".1. Fig 7 - the authors may consider indicating with a label what the difference is between A and C compared to B and D.1. L634 - why were only females used?1. L646 - "bran" should be "brain".1. L649 - "homoeothermic" should be "homeothermic".1. L661 - "allowed to generate" should be "allowed the generation of".1. L670 - no need for both "about" and "approximately".1. L681 - please state what the search stimuli were.1. L688 - should be "closed-field".1. L754 - add a hyphen to "time-consuming".

Our reply: Thanks so much for the detailed proofreading of the manuscript and suggestions. All them have been clarified or implemented and corrected in the text.

**Reviewer #2 (Recommendations For The Authors):**
The authors could investigate the effects of different doses of ACh on auditory processing to determine if there is a dose-response relationship.

Our reply: We agree that this is an interesting question also relate to a matter raised by Reviewer#1 that could be linked to the issue of ‘exogenous Ach’.

The study only investigated the effects of ACh on neuronal responses during an auditory oddball task. It would be interesting to investigate the effects of ACh on other aspects of auditory processing, such as sound localization or the discrimination of tones.

Our reply: We agree that, while these aspects of auditory processing are very fascinating, they were outside the scope of the study, and not directly related to predictive coding and precision, so each one of these characteristics would be a full, future project in itself.

The authors could provide more context on the significance of their findings for individuals with auditory processing disorders.

Our reply: Thanks for the suggestion. It remains unclear how abnormal brainstem and cortical processing associated with auditory processing disorders arises (Moore, 2006, 2012). While we are not aware of any known direct connection between auditory processing disorders and acetylcholine, individuals with auditory processing disorders do have difficulties with auditory selective attention, so perhaps one could speculate that ACh, by modulating SSA/prediction error, could have some impact on encoding salient events, and if disrupted could lead to problems with selective attention. Moore (2012) speculated that auditory processing disorders may arise from unbalanced processing in bottom-up and top-down contributions.

Since ACh has been implicated in some neurogenerative diseases and neurodevelopmental disorders, we have also added in the Discussion dialogue about a possible relationship between the modulatory effect of ACh on predictive coding (which involves bottom-up and top-down contributions) and auditory processing disorders. We also cite the recent work by Felix and colleagues (2019) which is the only study we have found on the effects of ACh on auditory processing disorders where they analyzed altered temporal processing at the level of the brainstem in α7-subunit of the nicotinic acetylcholine receptor (α7-nAChR)-deficient mice. After studying α7-nAChR knockout mice of both sexes and wild-type colony controls, they concluded that the malfunction of the CHRNA7 gene that encodes the α7-nAChR may contribute to degraded spike timing in the midbrain, which may underlie the observed timing delay in the ABR signals. These authors propose that their findings are consistent with a role for the α7-nAChR in types of neurodevelopmental and auditory processing disorders. There is also evidence on cholinergic system disfunction being related to the pathophysiology of Alzheimer’s disease (Pérez-González et al., 2022). For instance, disfunction of the synapses of cholinergic neurons in the hippocampus and nucleus basalis of Meynert, as well as decreased choline acetyltransferase activity, is associated to memory disorders in Alzheimer’s disease (Hampel et al., 2018). Also, A Alzheimer’s disease D patients show reduced amounts of the vesicular ACh transporter in some brain areas (Aghourian et al., 2017). Finally, cholinesterase inhibitors seem to have some favorable effect in the treatment of Alzheimer’s disease patients (Sharma, 2019).

Aghourian M, Legault-Denis C, Soucy J-P, Rosa-Neto P, Gauthier S, Kostikov A, et al. 2017. Quantification of brain cholinergic denervation in Alzheimer’s disease using PET imaging with [18F]-FEOBV. Mol. Psychiatry 22:1531–1538. doi: 10.1038/mp.2017.183

Felix RA 2nd, Chavez VA, Novicio DM, Morley BJ, Portfors CV. 2019. Nicotinic acetylcholine receptor subunit α7-knockout mice exhibit degraded auditory temporal processing. J Neurophysiol. 122(2):451-465. doi: 10.1152/jn.00170.2019.

Hampel H, Mesulam M-M, Cuello AC, Khachaturian AS, Vergallo A, Farlow MR, et al. 2018. Revisiting the Cholinergic Hypothesis in Alzheimer’s Disease: emerging Evidence from Translational and Clinical Research. J. Prev. Alzheimers Dis. 6:1–14. doi:10.14283/jpad.2018.43

Moore DR. 2006. Auditory processing disorder (APD)-potential contribution of mouse research. Brain Res. 1091:200–206.

Moore DR. 2012. Listening difficulties in children: bottom-up and top-down contributions. J Commun Disord. ;45:411–418.

Pérez-González D, Parras GG, Morado-Díaz CJ, Aedo-Sánchez C, Carbajal GV, Malmierca MS. 2021. Deviance detection in physiologically identified cell types in the rat auditory cortex. Hear Res. 2021 Jan;399:107997. doi: 10.1016/j.heares.2020.107997.

Pérez-González D, Schreiner TG, Llano DA and Malmierca MS. 2022. Alzheimer’s Disease, Hearing Loss, and Deviance Detection. Front. Neurosci. 16:879480. doi: 10.3389/fnins.2022.879480

Sharma K. 2019. Cholinesterase inhibitors as Alzheimer’s therapeutics. Mol. Med. Rep. 20:1479–1487. doi:10.3892/mmr.2019.1 0374